# Effects of combined nitrification inhibitors on soil nitrification, maize yield and nitrogen use efficiency in three agricultural soils

**Lei Cui[1,2], Dongpo Li[1]\*, Zhijie Wu[1], Yan Xue[1], Furong Xiao[1,2], Ping Gong[1], Lili Zhang[1], Yuchao Song[1], Chunxiao Yu[1], Yandi Du[3], Yonghua Li[4], Ye Zheng[5]**

**1** Institute of Applied Ecology, Chinese Academy of Sciences, Shenyang, Liaoning, China, **2** University of Chinese Academy of Sciences, Beijing, China, **3** Chaoyang County Agricultural Technology Extension Center, Chaoyang, Liaoning, China, **4** North Huajin Chemical Industries Group Corporation, Panjin, Liaoning, China, **5** Jinxi Natural Gas Chemical Co. Ltd, Huludao, Liaoning, China

☯ These authors contributed equally to this work.

\* lidp@iae.ac.cn

**Data Availability Statement:** All relevant data are within the paper.

**Funding:** This work was financially supported by the State Key Program of China

## Abstract

Application of nitrification inhibitors (NIs) with nitrogen (N) fertilizer is one of the most efficient ways to improve nitrogen use efficiency (NUE). To fully understand the efficiency of NIs with N fertilizer on soil nitrification, yield and NUE of maize (*Zea mays* L.), an outdoor pot experiment with different NIs in three soils with different pH was conducted. Five treatments were established: no fertilizer (Control); ammonium sulfate (AS); ammonium sulfate + 3, 4-dimethyl-pyrazolate phosphate (DMPP) (AD); ammonium sulfate + nitrogen protectant (N-GD) (AN); ammonium sulfate + 3, 4-dimethyl-pyrazolate phosphate + nitrogen protectant (ADN). The results showed that NIs treatments (AD, AN and ADN) significantly reduced soil nitrification in the brown and red soil, especially in AD and ADN, which decreased apparent nitrification rate by 28% - 44% ($P < 0.05$). All NIs treatments significantly increased yield and NUE of maize in three soils, especially ADN in the cinnamon soil and AD in the red soil were more efficiency, which significantly increased maize yield and apparent nitrogen recovery by 5.07 and 6.81 times, 4.39 and 8.16 times, respectively. No significant difference on maize yield was found in the brown soil, but AN significantly increased apparent nitrogen recovery by 70%. Given that the effect of NIs on both soil nitrification and NUE of maize, DMPP+N-GD was more efficient in the cinnamon soil, while N-GD and DMPP was the most efficiency in the brown and red soil, respectively. In addition, soil pH and soil organic matter play important role in the efficiency of NIs.

## Introduction

Nitrogen (N) fertilizers can increase food production by almost 50%. Hence, tons of N fertilizers were applied to obtain high yield of grain, which made a significant contribution in alleviating the global food shortage [1]. However, plants were rarely able to absorb more than 50% of the N fertilizer applied to cropping systems, and most of them were lost through $NO_3^-$

(2017YFD0200707), a grant to Dongpo Li, who designed the experiment.We are grateful to the National Field Research Station of Shenyang Agro-ecosystems, Chinese Academy of Sciences, for providing the experimental field.

**Competing interests:** We confirmed that this commercial affiliation does not alter your adherence to all PLOS ONE policies on sharing data and materials by including the following statement.

leaching, $NH_3$ volatilization and $N_2O$ emission due to the excessive application of N fertilizers, which caused many problems, such as lower nitrogen use efficiency (NUE), economical loss, and environment pollution [2]. It is necessary, therefore, to find out an efficient way to improve NUE, increase crop yield and mitigate environmental pollution.

Adding nitrification inhibitors (NIs) into ammonium-based fertilizer is one of the considerably effective technologies that can inhibit nitrification, reduce N loss, thus improving crop yield and NUE in agricultural systems [3,4]. Nitrification is a major process impacting N cycling in the high-production agricultural systems [5]. NIs are compounds that delay the process of nitrification of $NH_4^+$ to $NO_3^-$ and subsequent by depressing the activities of nitrifiers and denitrifiers in soil [6]. Nitrogen protectant (N-GD) is a mixture of DCD and DMPP in a certain proportion. Two commonly used NIs are 3,4-dimethylpyrazol phosphate (DMPP) and dicyandiamide (DCD). Many studies documented that the application of DMPP or DCD with organic fertilizer (cattle slurry, cow urine) and inorganic fertilizer (urea, ammonium chloride, ammonium sulfate) effectively reduced N loss and improved crop and grass yield and NUE in both agricultural and pasture system [7–11]. The application of DCD with urea significantly improved NUE by 12.9% and 14.6%, where reduced $NO_3^-$ leaching by 58.5% and 35.2% and $N_2O$ emission factor by 83.8% and 72.7% in two soils (Huangzongrang and chaotu soils), respectively [3]. Fan et al found that the application of NIs with urea increased ammonium concentrations by 0.3% - 41.1% and decreased nitrate concentrations by 6.3% - 34.4% [12]. However, there has been little research on the effect of NIs combined with ammonium sulfate in soils. It has been reported that sulfate ($SO_4^-$) is possibly reduced to thiosulfate ($S_2O_3^{2-}$), which is a nitrification inhibitor [13]. Moreover, the synergistic effect of anions and NIs might improve the inhibition effect of NIs [14]. In addition, DCD was more widely used in some countries especially in New Zealand as it was less volatile, easily souble in water and relatively cheaper than other NIs [15]. While DMPP has lower application of one-tenth of DCD dose, and it also has lower phytotoxicity to the plant [16,17]. DMPP is indiscriminately binding with ammonium monooxygenase, while DCD is blocking the electron transport in the cytochromes, which deactivate the enzyme responsible for the first step of nitrification [18]. Therefore, the present study addresses a combined application of different kinds of NIs to refine the use of NIs in soils.

The main factors affecting the effect of NIs are the properties of NIs, soil physicochemical properties (soil organic carbon (SOC), pH), fertilizer types (organic or inorganic), and field management [19,20]. It is worth to mention that soil pH is one of the main factors to influence the soil nitrification and the efficiency of NIs. In general, nitrification is easy in soils of pH $\geq$ 6.0, but not in soils of pH $\leq$ 5.0 [19]. A laboratory incubation experiment indicated that DMPP with ammonium chloride was more effective in inhibiting nitrification in the neutral soil (pH: 7.0; 93.5%) than in the alkaline soil (pH: 8.0; 85.1%) and acid soil (pH: 4.6; 70.5%) [21]. Another laboratory incubation experiment had shown that DMPP with urea maintained higher $NH_4^+$-N concentrations and lower $NO_3^-$-N concentrations, thus reducing N loss and improving NUE in the brown soil (pH: 6.31) [22]. While a field experiment study reported that DCD and urease inhibitor with urea had no significant difference in $NH_4^+$-N or $NO_3^-$-N in an acid soil, but in grain yield significantly higher than those of control treatment NIs, and no significant difference in grain yield was found between NIs and without NIs [23]. Another field experiment showed that DMPP significantly increased the annual crop yield by 6% relative to the urea treatment in an alkaline soil [24]. A meta—analysis showed that both DCD and DMPP were effective in increasing soil $NH_4^+$-N content combined with ammonium sulfate (AS), urea or organic fertilizer. DMPP was also effective in increasing $NH_4^+$-N content and decreasing $NO_3^-$N when combined with AS [25]. Many previous studies focused on the effect of NIs alone [26–28], and the responses of different NIs on yield and NUE are various due to the different soils. Therefore, it is urgent to fully understand the effect of different NIs on soil

nitrification, yield and NUE of maize in different soils to provide theoretical basis for choosing better NIs. In addition, this is the first research to study the effect of N-GD in different soils.

In this study, an outdoor pot experiment with different types of NIs additions in the above three soils was conducted. The objectives of this study were: 1) to examine the effect of NIs and their combination with ammonium sulfate on soil nitrification in three soils with different pH; 2) to compare the effect of NIs and their combinations in three soils with different pH; 3) to identify the effect of NIs on maize yield and NUE. We hypothesized that the combinations of NIs would be more efficient in three soils with different pH soil and soil pH may be the main factor affecting the effect of nitrification inhibitors.

## Materials and methods

### Study site and soils

An outdoor pot experiment was carried out at the National Field Observation and Research Station of Agroecosystems in Shenyang, Liaoning province (41˚31'N, 123˚24'E), in which Dongdan-6531 spring maize (*Zea mays* L., from May to October, 2018) was planted. The mean annual air temperature is 7–8˚C, and the mean annual precipitation is approximately 700 mm. The frost-free period is 147–164 days. The soils used in this study were obtained from three sites: a brown soil (Hap-Udic Luvisol in the FAO (Food and Agriculture Organization of the United Nations) WRB (World Reference Base) classification system) at Changtu country (42˚25'N, 123˚28'E), Liaoning Province of Northeast China, a cinnamon soil (Hap-Ustic Luvisol in the FAO WRB system) at Chaoyang City (41˚49'N, 122˚48'E), Liaoning Province of Northeast China, and a red soil (Haplic Luvisol in the FAO WRB system) at Yingtan National Agro-ecosystem Field Experiment Station (28˚15'N, 116˚55'E) of the Chinese Academy of Sciences in Jiangxi Province, China. Brown soil, cinnamon soil and red soil are the typical agricultural soils with different pH in China. The sampling sites were planted with maize and regularly fertilized. At each site, surface soil (0–20 cm) was collected, sieved to pass through a 5-mm mesh, and homogenized thoroughly. Soil physicochemical properties are shown in Table 1.

### Experimental design

Five treatments were established: 1) no N fertilizer (control); 2) ammonium sulfate (AS); 3) AS +3, 4-dimethyl-pyrazolate phosphate (DMPP) (AD); 4) AS+nitrogen protectant (N-GD) (AN); 5) AS+DMPP+N-GD (ADN). Each treatment had six replications. The fertilizer ammonium sulfate, triple superphosphate and potassium chloride were applied with an application

**Table 1. Physicochemical properties of the three agricultural soils (0–20 cm soil layers).**

| Soil property | Cinnamon soil | Brown soil | Red soil |
|---|---|---|---|
| pH | 7.90 ± 0.08 | 5.30 ± 0.02 | 4.70 ± 0.83 |
| Total C(g kg$^{-1}$) | 12.30 ± 0.20 | 9.24 ± 0.02 | 6.86 ± 0.35 |
| Total N(g kg$^{-1}$) | 1.07 ± 0.03 | 0.98 ± 0.08 | 0.89 ± 0.14 |
| NH$_4^+$-N(mg kg$^{-1}$) | 16.70 ± 0.13 | 13.83 ± 0.71 | 13.20 ± 0.83 |
| NO$_3^-$-N(mg kg$^{-1}$) | 21.28 ± 0.28 | 10.34 ± 0.25 | 11.53 ± 0.98 |
| Available P(mg kg$^{-1}$) | 5.70 ± 0.63 | 11.73 ± 1.04 | 13.70 ± 1.41 |
| Available K(mg kg$^{-1}$) | 91.32 ± 2.77 | 61.87 ± 2.89 | 131.49 ± 7.13 |
| SOM (g kg$^{-1}$) | 21.21 ± 0.34 | 15.94 ± 0.03 | 11.83 ± 0.60 |

Values indicate mean ± standard deviations (n = 3). Total C, total soil carbon; Total N, total soil nitrogen; NH$_4^+$-N, ammonium nitrogen; NO$_3^-$-N, nitrate nitrogen; P, phosphorus; K, potassium; SOM: Soil organic matter.

rate of 0.3g N, 0.12 g $P_2O_5$ and 0.15 g $K_2O$ per kg soil, respectively. The application rates of DMPP and N-GD were 0.5% and 1.5%, respectively on the w/w basis of N, and the application rate of every single NI was reduced by 50% in NIs combinations treatments. All the amendments were basal dressed, and air-dried soil (equivalent to 8 kg of oven-dry weight) was thoroughly mixed with the corresponding amendments before added a plastic pot of a 28-cm diameter with a 26-cm height. Soil moisture content was adjusted with deionized water to 60% of the maximum water-holding capacity (WHC), and watering was carried out every day during the maize growth period. Five seeds were sown in each pot, and the seedlings were thinned to one per pot after germination and seedling establishment. DMPP and N-GD were supplied by Zhejiang Chemical Institute, China and Spanish corporation, respectively.

## Sampling of soil and plant

Soil samples were collected at the seedling stage, elongation stage, filling stage, and maturity stage of maize, *i.e.* 30, 59, 95 and 135 days after planting, respectively, with three replicates for each treatment. For each pot, five soil cores were collected by using a soil auger (2.5 cm in diameter), and bulked into a composite sample. The samples were packed with ice pack and transported to laboratory, and passed through a 2 mm sieve before determining inorganic N ($NH_4^+$-N and $NO_3^-$-N) and moisture content.

The whole plant was harvested at the mature stage of maize from another three replications, and separated into straw and grain. The plant will be placed in oven, dried to constant weight at 65˚C to calculate aboveground biomass of maize, then ground to pass through a 0.25 mm sieve for the analysis of total nitrogen (TN).

## Analytical methods

Soil pH was tested in 1:2.5 soil-water ratio by using potentiometric method with a pH meter (METTLER TOLEDO S200, Shanghai, China), and TN of maize was determined by dry combustion using an elemental analyzer (Vario EL III, Germany) [29]. Soil available phosphorus (AP) was extracted with 0.5 mol $L^{-1}$ $NaHCO_3$ and analyzed by the molybdenum blue method [30], soil available potassium (AK) was extracted with 1 mol $L^{-1}$ $NH_4OAc$ and determined by the flame photometry method [30]. The soil $NH_4^+$-N and $NO_3^-$-N content were determined by extracting a 5-g soil subsamples with 50 ml of 2 mol $L^{-1}$ potassium chloride (KCl) and the samples were shaken for 1 h on a reciprocal shaker, filtered and the extract analyzed on a Continuous Flow Analyzer (AA III, Germany) [31].

## Calculations and statistical analysis

The soil apparent nitrification rate (ANR, %) is usually used to characterize the intensity of soil nitrification, which was calculated using Eq (1) [28]:

$$ANR = NO_3^- - N/(NH_4^+ - N + NO_3^- - N)*100 \tag{1}$$

Agronomic nitrogen use efficiency (ANUE, g $g^{-1}$) and apparent nitrogen recovery (AR, %) were calculated by Eq (2) and Eq (3):

$$ANUE = (Y - Y_C)/N_F \tag{2}$$

$$AR = (U - U_C)/N_F*100 \tag{3}$$

Where: Y, grain yield with N fertilizer; $Y_C$, grain yield with no fertilizer; $N_F$, the amount of N

fertilizer applied; U, plant nitrogen uptake in the aboveground parts with N fertilizer; $U_C$, plant nitrogen uptake in the above-ground parts with no N fertilizer [32].

A three -way ANOVA was used to analyse the impact of soil types (S), nitrification inhibitor (NI) and days after planting (D) and their interactions ((S*NI), (S*D), (NI*D) and (S*NI*D)) on the content of $NH_4^+$-N, $NO_3^-$-N and apparent nitrification rate (ANR). A two-way ANOVA was used to analyze the effects of soil types (S), nitrification inhibitor (NI) and their interactions (S*NI) on grain yield, ANUE and AR at the maturity stage of maize. Multiple comparisons among the treatments were further explained using Duncan test. Differences were considered significant at $P < 0.05$. All statistical analyses were performed using SPSS Version 22.0. Graphs were prepared with Origin 9.0.

## Results

### The contents of $NH_4^+$-N and $NO_3^-$-N in soils

The contents of $NH_4^+$-N and $NO_3^-$-N were significantly affected by soil types, nitrification inhibitor and days after planting (Table 2, $P < 0.05$). The application of N fertilizer and NIs significantly increased the soil inorganic N content in all three agricultural soils (Fig 1). However, there was no significant differences in soil $NH_4^+$-N content of all treatments at 135 days after maize planting, which decreased about 11 mg $kg^{-1}$, 10 mg $kg^{-1}$ and 24 mg $kg^{-1}$ in the cinnamon soil, brown soil and red soil, respectively (Fig 1A, 1C and 1E, $P > 0.05$). There was the same pattern between of $NO_3^-$-N content with the $NH_4^+$-N content, which increased at 30 days after maize planting and gradually declined followed the growth of maize (Fig 1B, 1D and 1F). In the cinnamon soil, the addition of NIs significantly increased soil inorganic N expect for AN. AD maintained higher $NH_4^+$-N content for a long time, while ADN had higher $NO_3^-$-N content during the growth of maize (Fig 1A, $P < 0.05$). In addition, ADN had higher inorganic nitrogen than AN (Fig 1B).

In the brown soil, the $NH_4^+$-N content in AD was the highest than that in other treatments at30 days after maize planting (Fig 1C, $P < 0.05$). No significant differences were detected between AN and ADN for $NH_4^+$-N content (Fig 1C, $P > 0.05$). At 59 days after maize planting, the $NH_4^+$-N content in AD and AN was higher compared with ADN (Fig 1C). AN also

**Table 2. Three-way ANOVA ($P < 0.05$) of soil types (S), nitrification inhibitor (NI) and days after planting (D) and their interactions ((S*NI), (S*D), (NI*D) and (S*NI*D)) on the content of NH4+-N, NO3—N and apparent nitrification rate (ANR).**

| Factors | DF | NH4+-N | | | NO3--N | | | ANR | | |
|---|---|---|---|---|---|---|---|---|---|---|
| | | SS | F | p | SS | F | P | SS | F | p |
| S | 2 | 13347.4 | 4094.4 | *** | 1037.9 | 1540.3 | *** | 14333.3 | 1465.1 | *** |
| NI | 3 | 1073.1 | 219.5 | *** | 49.6 | 49.0 | *** | 74.5 | 5.1 | ** |
| D | 3 | 30164.2 | 6168.7 | *** | 9501.7 | 9400.4 | *** | 8221.6 | 560.3 | *** |
| S*NI | 6 | 1088.5 | 111.3 | *** | 283.2 | 140.1 | *** | 874.9 | 29.8 | *** |
| S*D | 6 | 9895.4 | 1011.8 | *** | 665.2 | 329.0 | *** | 3300.3 | 112.4 | *** |
| NI*D | 9 | 1555.7 | 106.1 | *** | 190.4 | 62.8 | *** | 700.0 | 15.9 | *** |
| S*NI*D | 18 | 1243.7 | 42.4 | *** | 734.3 | 121.1 | *** | 1341.6 | 15.2 | *** |
| Model | 47 | 58368.1 | 761.9 | *** | 12462.2 | 787.0 | *** | 28846.2 | 125.5 | *** |
| Error | 96 | 156.5 | | | 32.3 | | | 469.6 | | |

SS, the sum of squares.

F value, the ratio of mean squares of two independents samples.

*** Indicates significance at $P < 0.001$

** Indicates significance at $P < 0.01$.

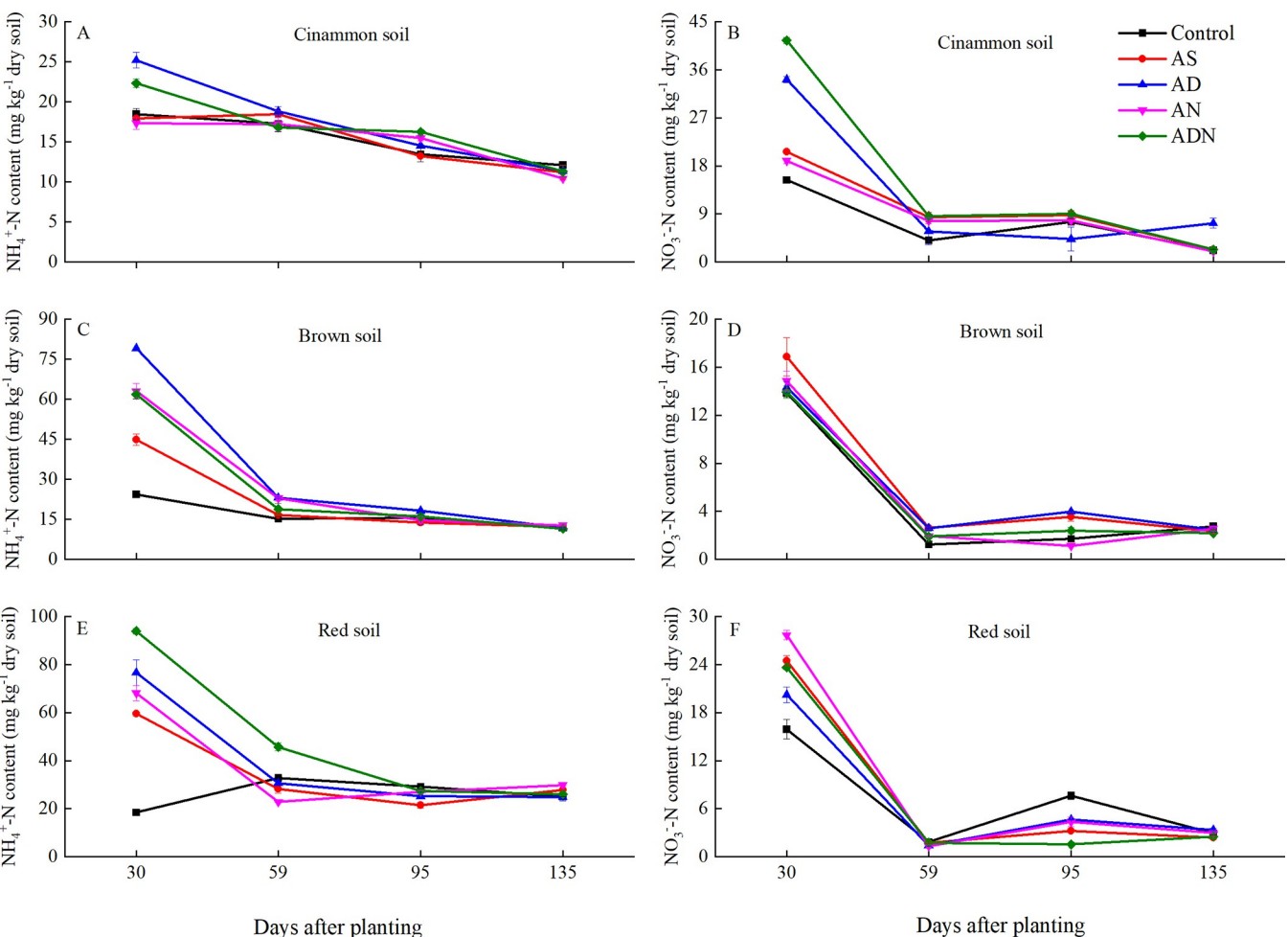

**Fig 1. Dynamic changes of NH4+-N and NO3−-N of different treatments in three soils.** Treatment: Control, no fertilizer and nitrification inhibitors; AS, ammonium sulphate; AD, AS+3, 4-dimethylpyrazole phosphate (DMPP); AN, AS+nitrogen protectant (N-GD); ADN, AS+3, 4-dimethylpyrazole phosphate (DMPP)+nitrogen protectant (N-GD). Error bars represented standard deviations ($n = 3$).

maintained higher $NH_4^+$-N content until the 135 days after maize planting. All treatments with NIs significantly decreased $NO_3^-$-N content at 30 days after maize planting, especially AN was lower in $NO_3^-$-N content at the later stages (Fig 1D). Moreover, AN and ADN maintained higher nitrification inhibition during the four sampling periods (S1 Table).

In the red soil, all treatments with NIs significantly increased $NH_4^+$-N content compared with AS at 30 days after maize planting (Fig 1E, $P < 0.05$). In addition, ADN maintained higher $NH_4^+$-N content for a long period. Lower $NO_3^-$-N values in AD and ADN were found compared with AS, while ADN had no significant difference in $NO_3^-$-N content with AS (Fig 1F, $P > 0.05$).

## Apparent nitrification rate

Apparent nitrification rate (ANR) indicates the intensity of soil nitrification process. The lower the value is, the weaker the inhibition intensity of nitrification inhibitors on soil nitrification process is; the higher the value is, the higher the nitrification process intensity is. Soil types, nitrification inhibitor and days after planting significantly affected ANR (Table 2, $P < 0.05$). ANR was higher in the cinnamon soil than that in the other two soils (Fig 2). In the

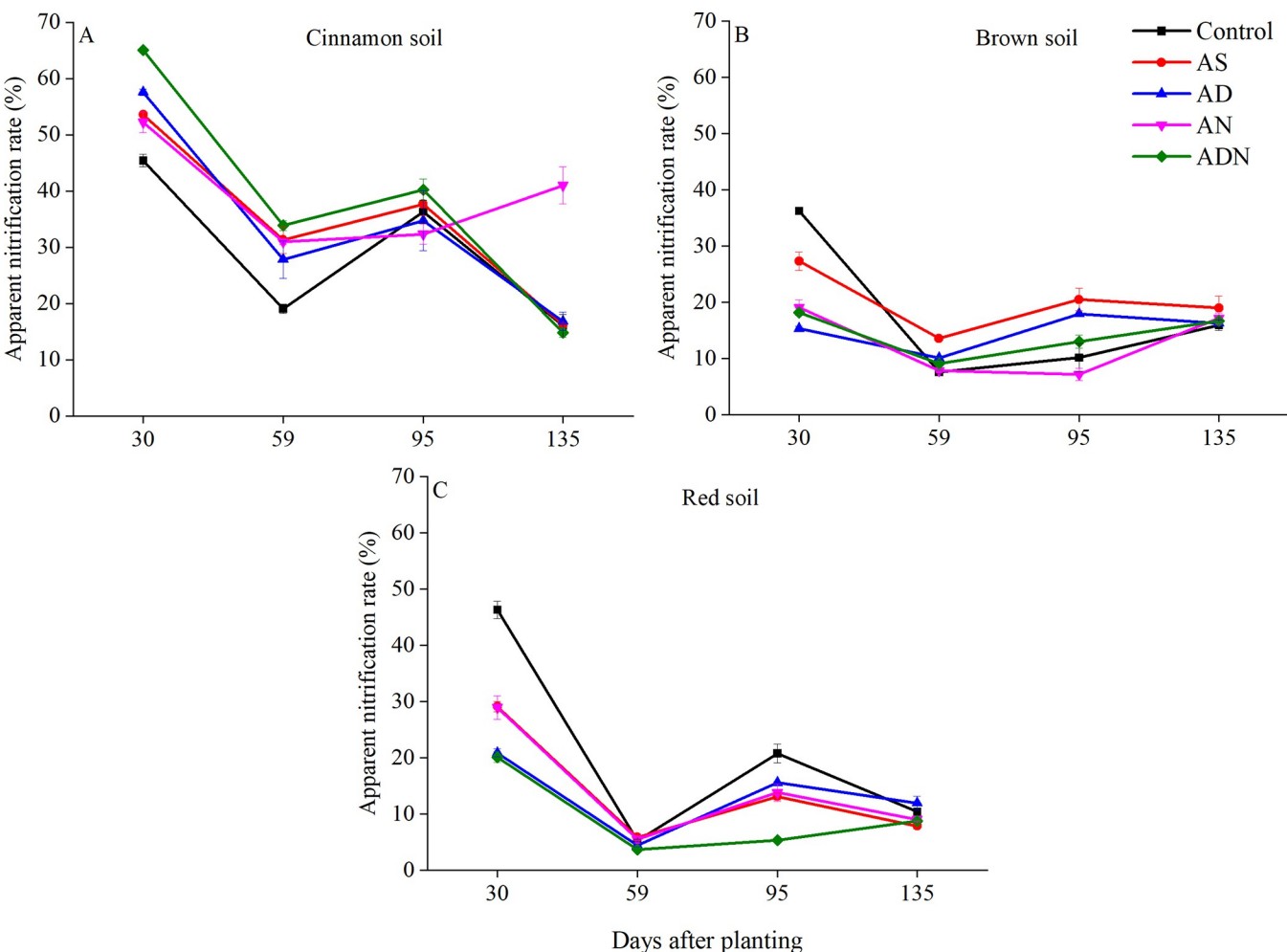

**Fig 2. Changes of apparent nitrification rate of three soils during four sampling periods.** Treatment: AS, ammonium sulphate; AD, AS+3, 4-dimethylpyrazole phosphate (DMPP); AN, AS+nitrogen protectant (N-GD); ADN, AS+3, 4-dimethylpyrazole phosphate (DMPP)+nitrogen protectant (N-GD). Error bars represented standard deviations ($n$ = 3).

cinnamon soil, higher ANR was observed in treatments of adding NIs into N fertilizer, especially in ADN, which showed that NIs significantly increased soil inorganic nitrogen (Fig 2A). In the brown soil, all treatments with NIs significantly reduced ANR over AS during the whole stage of maize growth. AD was the lowest at 30 days after maize planting, followed by AN (Fig 2B). In the red soil, the lowest values were observed in both AD and ADN (Fig 2C, $P < 0.05$). In addition, ADN significantly inhibited nitrification during the whole growth stage of maize (Fig 2C).

## Aboveground biomass and nitrogen uptake of maize

**Aboveground biomass.** In general, NIs with ammonium sulfate significantly increased aboveground biomass compared to AS in all the tested soils (Fig 3, $P < 0.05$). Grain yield of maize was significantly affected by both soil types and nitrification inhibitor (Table 3, $P < 0.01$). Both grain and straw biomass had the highest value in the brown soil than those in the cinnamon soil and red soil (Fig 3). In the cinnamon soil, the highest value of grain yield was found in ADN than that in other treatments (Fig 3A, $P < 0.05$). AN resulted in significant

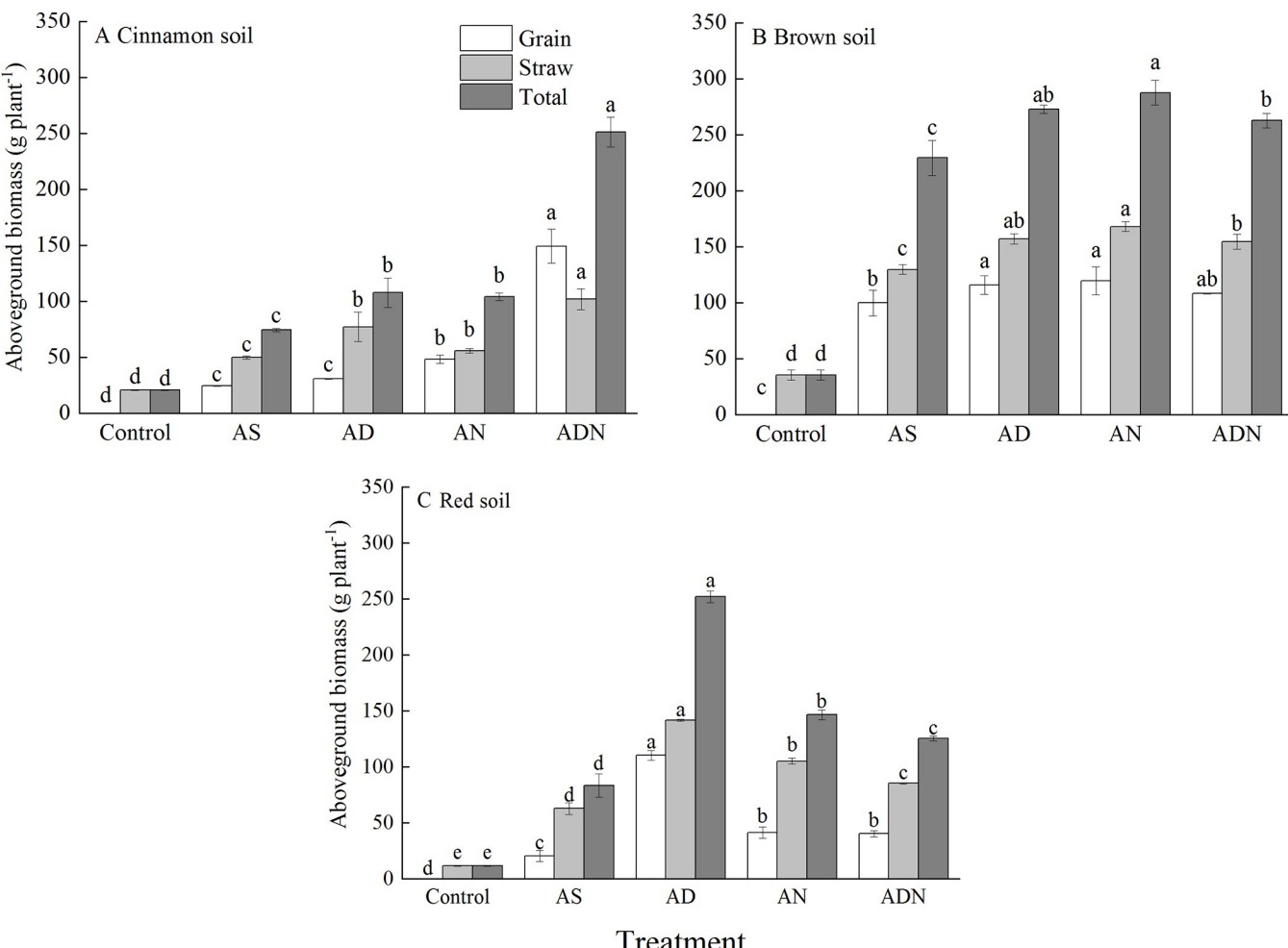

**Fig 3. Aboveground biomass of maize under different treatments in three soils.** Treatment: AS, ammonium sulphate; AD, AS+3, 4-dimethylpyrazole phosphate (DMPP); AN, AS+nitrogen protectant (N-GD); ADN, AS+3, 4-dimethylpyrazole phosphate (DMPP)+nitrogen protectant (N-GD). Error bars represented standard deviations (*n* = 3). Different letters indicate significant differences between different treatments at *P* < 0.05 by Duncan test.

increase in grain yield when compared to AS (*P* < 0.05), while there was no significant difference between AD and AS for grain yield of maize (*P* > 0.05).

In the brown soil, grain yield was the highest in AN followed by AD and AN, which were significantly higher than AS (Fig 3B, *P* < 0.05). No significant differences were detected in the application of NIs with ammonium sulfate (Fig 3B, *P* > 0.05).

In the red soil, the grain yield in NIs treatments was higher compared with AS (Fig 3C, *P* < 0.05). The highest value was found in AD in comparison to AN and ADN (Fig 3C).

**Nitrogen uptake of maize.** In the three agricultural soils, all treatments with NIs significantly increased N uptake by maize compared with control (Fig 4, *P* < 0.05). The highest value of grain and total nitrogen uptake of maize was found in the cinnamon soil (Fig 4). In the cinnamon soil, the highest N uptake of maize value was detected in ADN, which was significantly higher than other treatments (Fig 4A, *P* < 0.05). AN had higher N uptake of maize than that of AS, while AD had no significant difference from AS (Fig 4A, *P* < 0.05).

In the brown soil, there was no significant difference between AD and AN, which were significantly higher in N uptake of maize than other treatments (Fig 4B, *P* < 0.05). N uptake of maize from ADN was no significantly different from AS (Fig 4B, *P* > 0.05).

**Table 3. Two-way ANOVA ($P < 0.05$) of soil types (S) and nitrification inhibitor (NI) on grain yield, agronomic nitrogen use efficiency (ANUE) and apparent nitrogen recovery (AR) of maize.**

| Factors | DF | Grain yield | | | ANUE | | | AR | | |
|---|---|---|---|---|---|---|---|---|---|---|
| | | SS | F | p | SS | F | p | SS | F | p |
| S | 2 | 22787.7 | 201.8 | *** | 3955.7 | 201.8 | *** | 1274.2 | 31.8 | *** |
| NI | 3 | 12958.1 | 76.5 | *** | 2250.0 | 76.5 | *** | 7682.6 | 127.8 | *** |
| S*NI | 6 | 32174.4 | 95.0 | *** | 5585.7 | 95.0 | *** | 13024.2 | 108.4 | *** |
| Model | 11 | 67920.2 | 109.3 | *** | 11791.4 | 109.3 | *** | 21981.0 | 99.8 | *** |
| Error | 24 | 1355.4 | | | 235.3 | | | 480.8 | | |

SS, the sum of squares.

F value, the ratio of mean squares of two independents samples.

*** Indicates significance at $P < 0.001$.

In the red soil, NIs with N fertilizer significantly increased N uptake of maize compared with AS. AD significantly increased N uptake of maize compared with other treatments (Fig 4C, $P < 0.05$), while ADN was better than AN (Fig 4C).

### Agronomic nitrogen use efficiency and recovery of apparent nitrogen

Agronomic nitrogen use efficiency (ANUE) and apparent nitrogen recovery (AR) are different aspects to explain the utilization of nitrogen fertilizer in maize. Soil types and nitrification inhibitor significantly affected ANUE and AR (Table 3, $P < 0.05$). ANUE was the highest in the brown soil, while the highest value of AR was found in the red soil (Fig 5). In the cinnamon soil, the highest values in both ANUE and AR were from ADN, which was significantly different from other treatments. AN was higher in NUE than AD and AS. AD in AR was significantly higher than AS, while ANUE from AD had no significant differences from AS treatment (Fig 5A, $P > 0.05$).

In the brown soil, all treatments with NIs significantly improved ANUE and AR than those of AS (Fig 5B, $P < 0.05$), while there were no significant differences among them in ANUE. AR in AD and AN was significantly higher compared with ADN and AS. No significant differences in AR were detected between ADN and AS (Fig 5B, $P > 0.05$).

In the red soil, AD had the highest value in both ANUE and AR among all treatments, which was significantly different with the other treatments (Fig 5C, $P < 0.05$). No significant differences in ANUE were found between AN and ADN (Fig 5C, $P > 0.05$), while higher value in AR was found in ADN than AN.

## Discussion

### Effects of NIs on soil nitrification

The application of NIs have the potential to slow soil nitrification, thus increasing $NH_4^+$-N and decreasing $NO_3^-$-N content, and improving crop yield, aboveground biomass, N uptake and NUE. It is worth to mention that nitrification in the red soil has become stronger after tillage in recent decades, which is according with Lu et al [33]. In the present study, NIs treatments considerably reduced soil nitrification for all three soils, especially in both brown soil and red soil (S1 Table), a result which was mainly driven by lower pH and fertility. The efficacy of NIs in inhibiting nitrification was reduced because of the rapid hydrolysis of NIs at high soil pH [34].

The effect of NIs varied among different types of soil. NIs significantly increased soil $NH_4^+$ in both soils expect for cinnamon soil, which is consistent with Guo et al [35], who found that

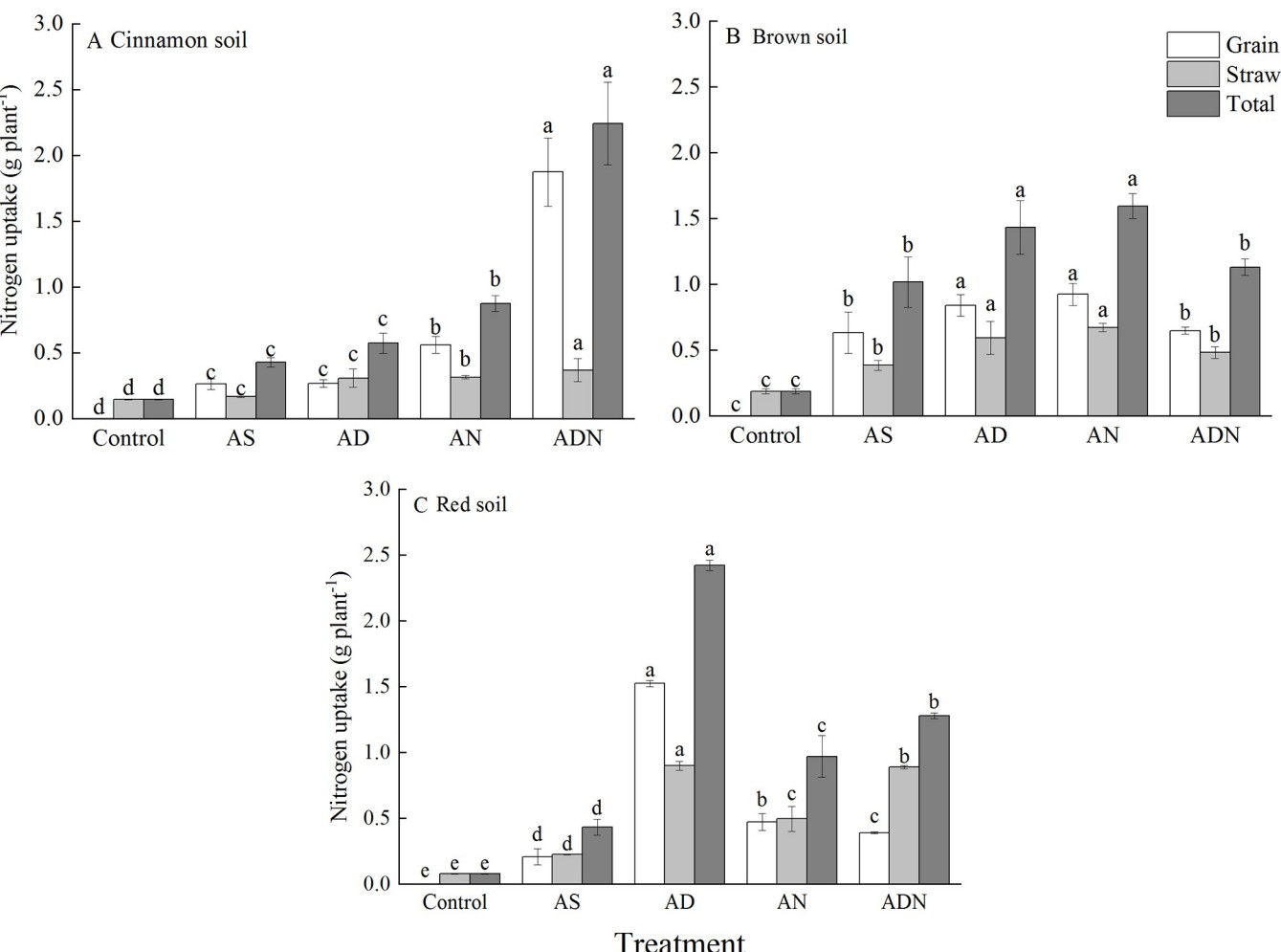

**Fig 4. Nitrogen uptake of maize under different treatments in three soils.** Treatment: AS, ammonium sulphate; AD, AS+3, 4-dimethylpyrazole phosphate (DMPP); AN, AS+nitrogen protectant (N-GD); ADN, AS+3, 4-dimethylpyrazole phosphate (DMPP)+nitrogen protectant (N-GD). Error bars represented standard deviations ($n$ = 3). Different letters indicate significant differences between different treatments at $P < 0.05$ by Duncan test.

DCD with urine had the potential to increase $NH_4^+$ content and maintained it higher for a long time. However, NIs reduced ANR in all three soils, which is in line with Gong et al [36], who found that DCD and DMPP with urea significantly suppressed potential nitrification rate. DMPP significantly inhibited nitrification in all three soils. DMPP is indiscriminately binding and interaction with ammonium monooxygenase to inhibit the first rate-limiting step of soil nitrification [37]. In addition, DMPP is a heterocyclic N compound with the advantages of low mobility, slow biodegradation and persistence [38]. At the later stage of maize, $NH_4^+$-N content declined gradually, which is in line with Zaman et al [9]. The main reasons were the soil nitrification (Fig 1) (decomposition of NIs at the later stage of maize) [9], N uptake of maize (Fig 4) [36] and microbial immobilization [39,40].

### Effects of NIs on maize yield and NUE

There are many studies to test the effect of NIs on crops production and N uptake, but it is difficult to draw general conclusions because the performance of the options varied across sites due to soil type (pH, SOM), NIs type, the forms of N fertilizer (organic N fertilizer (urine),

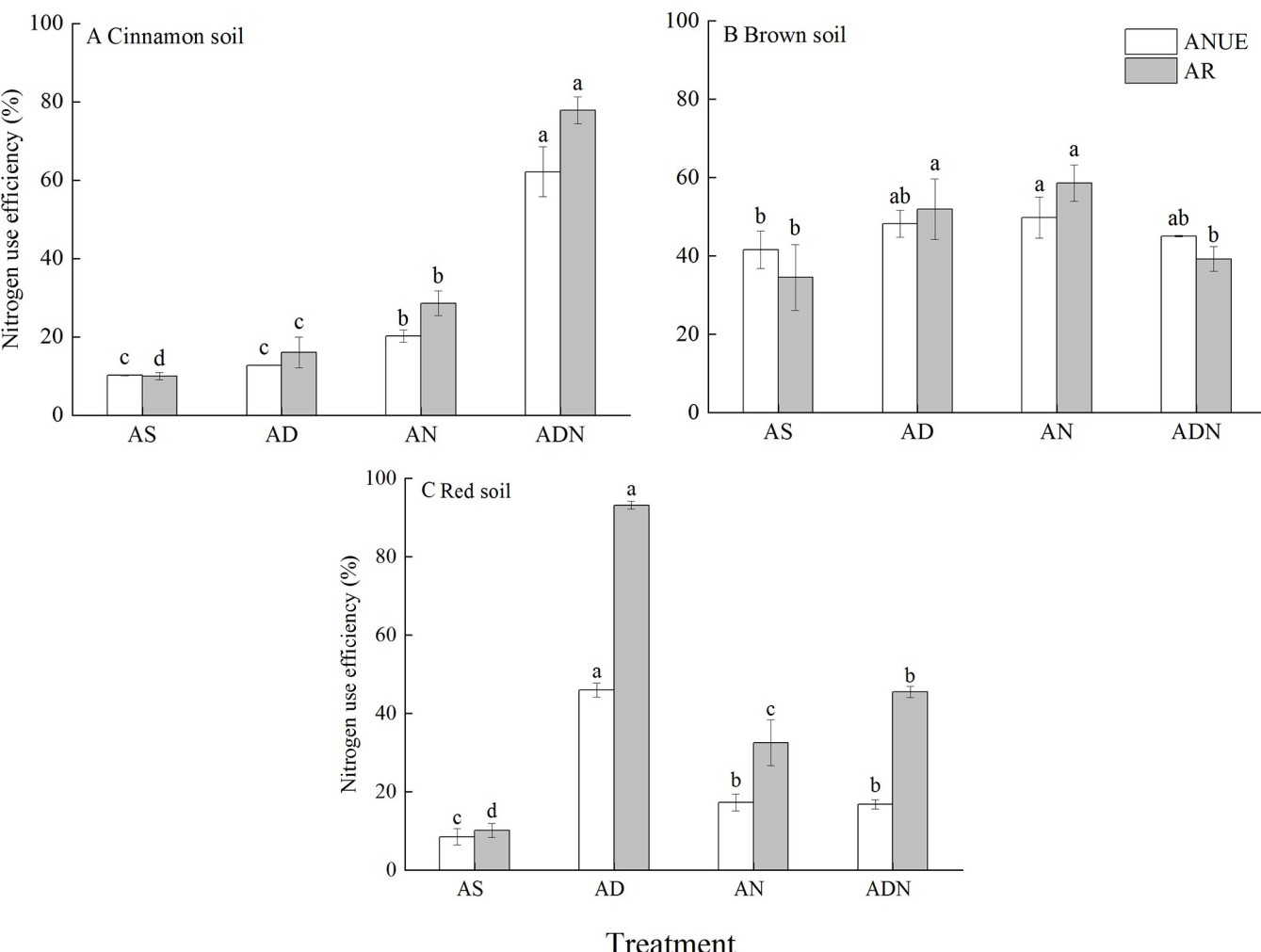

**Fig 5. Nitrogen use efficiency (ANUE and AR) under different treatments in three soils.** ANUE, agronomic nitrogen use efficiency; AR: Apparent nitrogen recovery. Treatment: AS, ammonium sulphate; AD, AS+3, 4-dimethylpyrazole phosphate (DMPP); AN, AS+nitrogen protectant (N-GD); ADN, AS+3, 4-dimethylpyrazole phosphate (DMPP)+nitrogen protectant (N-GD). Error bars represented standard deviations ($n$ = 3). Different letters indicate significant differences between different treatments at $P < 0.05$ by Duncan test.

inorganic N fertilizer (urea, ammonium sulfate et al) and so on [18]. In the current research, NIs significantly increased yield and N uptake of maize compared with the application of ammonium sulfate among the three soils, especially DMPP and N-GD in the brown soil, DMPP+N-GD in the cinnamon soil and DMPP in the red soil. Cui et al also found that DCD with urea significantly increased the yield of radish and N concentrations by 10.4%–36.2% in vegetables [3]. A field experiment also showed that the application of DMPP increased the grain yield of rice and rape by 4.2%–4.7% and 6.6%–7.5%, respectively [41], which is in line with the present study. NIs can maintain higher $NH_4^+$-N content for a longer period, which improve crop more available N absorption [42]. Ammonium ions are usually adsorbed by clay particles or soil organic matter, thus decreasing N loss [25]. Moreover, NIs with N fertilizer can increase the synchrony between N supplied and crop demand [43]. However, the results of Huérfano et al and Guardia et al showed that there was no any effect of NIs on maize yield [44,45], which is in contrast to the results. The possible explanation is the effect of NIs mainly due to environmental factors (pot, field) and the type of fertilizer (ammonia sulfate, ammonia

sulphate nitrate, calcium ammonium nitrate) [46]. The higher values of plant N uptake imply higher NUE for NIs. Obviously, NIs also significantly improved NUE in this study. The results are agreement with He et al [47], which also found NIs with urea increased NUE.

In terms of cost, the price of DMPP is so expensive that its wide application is limited [25]. In the present study, DMPP had the similar efficiency with N-GD in the brown soil, while in the cinnamon soil DMPP+N-GD was more effective than DMPP alone, thus the best choice in the brown soil and cinnamon soil is the application of N-GD and DMPP+N-GD, respectively.

### Effects of soil properties on NIs use

The efficiency of NIs differed in different agricultural soils may be attributed to the differences in physicochemical properties of three tested soils, especially soil pH and SOM. The results of Pearson correlation revealed that soil pH and SOM were negatively correlated with soil $NH_4^+$-N and $NO_3^-$-N contents (Table 4, $P < 0.05$). The results confirmed our hypothesis that soil pH was the main factor influenccing the effect of nitrification inhibitors. Soil pH has been considered as one of the most important factors affecting the availability of NIs, because pH has potential to impact the mobility and degradation rate of the NIs in soils [48]. A meta—analysis showed that acidic soils (pH ≤ 6) showed a higher positive response to inhibitor application than neutral (pH 6–8) and alkaline soils (pH ≥ 8) [49]. The efficiency of DMPP was more stable in lower pH soil than that in alkaline soil. DMPP also performed better in neutral than in alkaline soil [21], which is similar to our results. It may be due to that the microbial activity in soils with higher pH is generally higher, which will accelerate the degradation rate of nitrification inhibitors. Additionally, the ANR in lower pH soils was lower than in alkaline soils, which is in accordance with the result of Lu et al [32]. In addition to soil pH, SOM also affected the effectiveness of NIs. SOM can absorb NIs and provide energy source for the microorganisms, which leads to the degradation of NIs, decreases the ability of NIs to inhibit nitrification [50]. The results of this paper are similarly to that of the literature, SOM was higher in the cinnamon soil than that in both brown and red soil, so the efficiency of DMPP in the cinnamon soil was lower than in the other two soils. While the effect of DMPP+N-GD had little influenced by SOM.

In addition, there is less study of N-GD, so more research should be required to understand its effect and mechanism. Moreover, the effect of NIs on the N cycle in a range of soil types, cultivated vegetation types, and climatic condition should be fully elucidated.

## Conclusions

Nitrification inhibitors (NIs) with ammonium sulfate can significantly influence soil inorganic nitrogen, improve yield and nitrogen use efficiency (NUE) of maize. NIs combination 3, 4-dimethyl-pyrazolate phosphate (DMPP) + nitrogen protectant (N-GD) was the best way to increase soil available nitrogen and improve NUE in the cinnamon soil, while N-GD was more

**Table 4. Pearson correlation between soil pH, soil organic matter (SOM), NH4+-N and NO3—N.**

| Item | $NH_4^+$-N | $NO_3^-$-N |
|---|---|---|
| **pH** | -0.175 | -0.563** |
| **SOM** | -0.415* | -0.362* |
| **$NH_4^+$-N** | 1 | -0.518** |

* Correlation is significant at the 0.05 level (two-tailed)

** Correlation is significant at the 0.01 level (two-tailed).

efficient in decreasing soil nitrification, improving yield and NUE of maize in the brown soil, DMPP was the most efficient in lowering soil nitrification and increasing maize yield and NUE in the red soil. Additionally, soil pH and soil organic matter are the main factors affecting the efficiency of NIs.

## Supporting information

**S1 Table. Nitrification inhibition rate of different treatments in three agricultural soils during four sampling periods.**
(DOCX)

## Acknowledgments

We are grateful to the National Field Research Station of Shenyang Agroecosystems, Chinese Academy of Sciences, for providing the experimental field.

## Author Contributions

**Conceptualization:** Lei Cui, Dongpo Li.

**Data curation:** Lei Cui.

**Formal analysis:** Lei Cui.

**Funding acquisition:** Dongpo Li, Zhijie Wu.

**Investigation:** Lei Cui.

**Methodology:** Lei Cui.

**Project administration:** Zhijie Wu, Lili Zhang.

**Resources:** Yan Xue, Furong Xiao, Ping Gong, Lili Zhang, Yuchao Song, Chunxiao Yu, Yandi Du, Yonghua Li, Ye Zheng.

**Software:** Lei Cui.

**Supervision:** Lei Cui, Dongpo Li, Zhijie Wu.

**Validation:** Lei Cui.

**Visualization:** Lei Cui.

**Writing – original draft:** Lei Cui.

**Writing – review & editing:** Lei Cui, Dongpo Li, Ping Gong, Lili Zhang.

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
