## [Decision Letter · Decision Letter 0]

16 Jul 2021

PONE-D-21-09541

Effects of combined nitrification inhibitors on soil nitrification, maize yield and nitrogen use efficiency in three agricultural soils

PLOS ONE

Dear Dr. Li,

Thank you for submitting your manuscript to PLOS ONE. After careful consideration, we feel that it has merit but does not fully meet PLOS ONE’s publication criteria as it currently stands. Therefore, we invite you to submit a revised version of the manuscript that addresses the points raised during the review process.

Please consider carefully all the considerations done by the reviewers. I think the manuscript should be greatly improved if doing so!

We look forward to receiving your revised manuscript.

Kind regards,

Manuel Joaquín Reigosa, Ph.D.

Academic Editor

PLOS ONE

Journal Requirements:

We note that one or more of the authors are employed by a commercial company: North Huajin Chemical Industries Group Corporation, Jinxi Natural Gas Chemical Co. Ltd.

2.1. Please provide an amended Funding Statement declaring this commercial affiliation, as well as a statement regarding the Role of Funders in your study. If the funding organization did not play a role in the study design, data collection and analysis, decision to publish, or preparation of the manuscript and only provided financial support in the form of authors' salaries and/or research materials, please review your statements relating to the author contributions, and ensure you have specifically and accurately indicated the role(s) that these authors had in your study. You can update author roles in the Author Contributions section of the online submission form.

2.2. Please also provide an updated Competing Interests Statement declaring this commercial affiliation along with any other relevant declarations relating to employment, consultancy, patents, products in development, or marketed products, etc.  

Reviewers' comments:

Reviewer's Responses to Questions

**Comments to the Author**

1. Is the manuscript technically sound, and do the data support the conclusions?

Reviewer #1: Yes

Reviewer #2: Yes

Reviewer #3: Yes

Reviewer #4: Yes

2. Has the statistical analysis been performed appropriately and rigorously? 

Reviewer #1: Yes

Reviewer #2: Yes

Reviewer #3: Yes

Reviewer #4: Yes

3. Have the authors made all data underlying the findings in their manuscript fully available?

Reviewer #1: Yes

Reviewer #2: Yes

Reviewer #3: Yes

Reviewer #4: Yes

4. Is the manuscript presented in an intelligible fashion and written in standard English?

Reviewer #1: Yes

Reviewer #2: Yes

Reviewer #3: Yes

Reviewer #4: No

5. Review Comments to the Author

Reviewer #1: This manuscript described the effects of different nitrification inhibitors on soil nitrification and maize yield in various agricultural soils with different physicochemical properties. The subject is interesting and the paper is well organized. The introduction gave a satisfactory literature survey. The research methods used in this paper were described in detail and the data was well statistically analyzed. Appropriate figures were given to make the paper understood easily.

From my point of view, the work is well-done and provides theoretical basis to utilize fertilizer nitrogen effectively on maize in three soils with different nitrification inhibitors. The paper conclusions provided critical support for the theory that different nitrification inhibitors have different effects in different physical and chemical properties of soils. In addition, it seeks to find an economical way to improve NUE with new nitrification inhibitor N-GD due to higher price of DMPP. The results also showed that soil pH plays an important role in the effectiveness of DMPP, N-GD and their combination in influencing soil nitrification, yield and NUE of maize. Therefore, this article deserves to be published.

Some minor modifications should be made before publication:

1. In this manuscript, "decrease nitrification" should be revised as “inhibit/suppress nitrification".

2. Page 2, line 24, please revise “The application” to “ Application”.

3. Page 2, line 25, please revise “the most efficiency” to “the most efficient”.

4. Page 2, line 38, please add “that” after “than”.

5. Page 3, line 59, please delete “which”.

6. Page 4, line 78, please add “and” in front of “field”.

7. Page 4, line 82, please add parentheses.

8. Page 5, line 90, please delete “are”.

9. Page 7, line 139, please revise “sampling” to “samples”.

10. Page 11, line 216, Page 16, line 332, please revise “was” to “were”.

11. Page 17, line 341, please add “of maize” after “NUE”.

12. Page 17, line 343, “improving” should be revised to “improve”.

Reviewer #2: To understand the efficiency of nitrification inhibitors with N fertilizer on soil nitrification, yield and nitrogen use efficiency of maize (Zea mays L.), an outdoor pot experiment with different nitrification inhibitors in three soils with different pH were conducted in the paper. The reviewer think that the problems of this paper are as follows:

1.There are some sentences with irregular grammar or incomplete meaning: Such as:

(1)L39:”DMPP and N-GD were more effective in the red soil than in the cinnamon soil”What is more effective?

(2)L40 and 41:“… than other two soils, … than other two soils” were changed to“…than that in other two soils,… than that in other two soils.”

(3)L169:“NH4+-N content and NO3--N content” should be changed to“Contents of NH4+-N and NO3—N in soils”.

(4)L197:“Apparent nitrification rate (ANR) indicates that intensify of soil nitrification”, the sentence is not a clause, it should be changed to “Apparent nitrification rate (ANR) indicates the intensify of soil nitrification” .

(5)L253:“Agronomy nitrogen use efficiency and apparent nitrogen recovery” should be changed to“Utilization efficiency of agronomy ang recovery of apparent nitrogen”.

(6) L327:“was more stable in lower pH soil than in alkaline soil” should be changed to“was more stable in lower pH soil than that in alkaline soil”.

2.Improper expression or incomplete meaning of some section titles:

(1)L146:“Analytical procedures” should be changed to “Analytical methods” (Only the determination methods are given, and there are no determination procedures or operation processes)

(2)Note: To distinguish the relationship between the instrument and the method, the instrument only is the means of the method, but the instrument itself is not the method. Such as: “with a pH meter” in L147, “with an elemental analyzer” in L148, “determined by the flame photometer method” in L150-151, and “using a Continuous Flow Analyzer” in L151-152. Above all, these sentences should be expressed in the form of “methods” rather than “instruments”, especially, “determined by the flame photometer method” in L150-151 was a wrong expression, should be changed to “determined by the flame photometry method”.

(3) The meaning of “Apparent nitrification rate”in L196 was too general.

3.Some tense usageswere not correct,the past tense should be used in many sentences but used the present tense.

4.In the description of result and content discussion, the first person is used too much. Such as: "we conduct an outdoor.." in L99, “We clearly observed that…..” in L170(Figure 1 should be described first), “We also found that…” in L179, “we clearly observed……” in L340(It is more appropriate to have such a statement in the conclusion). It is suggested that the result or content described in the first person should be described in the passive voice.

5.“In our study” is used too often, which is to discuss the results of this paper, so it is unnecessary to emphasize it many times.

6. The expression of this comparison "The results was similarly to our study,....."in L332 is improper, it should be expressed "The results of this paper was similarly to that of the literature,.....".

7.The part of “conclusion” is an objective statement of the law presented by the research results, The expression of “In our study, we clearly observed” should be deleted.

8.The logical relationship between several experimental results in the result part is not very clear, moreover, the logical relationship between the results and the discussion is also fuzzy.

Reviewer #3: Opinion on manuscript: PONE-D-21-09541 major revision

Global food security is a key challenge of the modern world. The demand for food is growing exponentially and in this regard agricultural production has to be increased to meet global food demands. One of the key agricultural practices is the higher application of nitrogenous fertilizers. However, the use efficiency of applied fertilizers is currently very low. This low efficiency is attributed to the great losses of nitrogenous fertilizers after their application into the soil due to leaching, run off, volatilization and de-nitrification. Overall, the findings of this study have essential implications for the enhancing NUE and apparent nitrogen recovery. The comments are as follows, and most of them are suggestions to the authors that should be taken on board for a revised version.

Specific comments:

1. Page 2 - line 25 –“efficiency ways” should be changed into “efficient ways”

2. Page 2 - line 32 –“which decreased apparent nitrification rate 28%” should be changed into “which decreased apparent nitrification rate by 28%”

3. Page 2 - line 35 –“5.07 times” delete “times”

4. Page 4 - line 67,69, 88 – “But” “And” should not be the first word of the sentence and rewrite this.

5. Page 4 - line 81 – “A laboratory incubation experiment…” should be marked with a reference.

6. Page 8 - line 147 – “pH meter” should be marked with the mode.

7. Page 8 – line 149 – “molybdenum blue method” reference?

8. Page 10 - line 197 – “Apparent nitrification rate” should be abbreviated.

9. Page 13 - line 255 – “Agronomy nitrogen use efficiency” should be changed into “agronomic nitrogen use efficiency”.

10. Page 14 - line 289 – “which was line in with” Grammatical errors

10. Page 15 - line 295-297 the discussion is so simple, rewrite this part.

10. Page 14 - line 314 – “environmental factors, soil properties and the type of fertilizer” should be more detailed. Rewrite this part according to the results of this article.

10. Page 16- line 323 – “In our study, the efficiency of NIs differed in different agricultural soils, mainly due to the different pH”. The conclusion is far fetched and rewrite this sentence.

The effects of pH on nitrification should be more detailed.

11. Page 16- line 322 – Please add a “Table for correlation analysis between soil properties and NIs use” if possible.

12. please calculate the nitrification inhibition rate of NIs

13. The whole manuscript some of the expressions are too colloquial and the meaning is not clearly expressed, please check the grammar in the article carefully.

Reviewer #4: This manuscript had compared the effects of combined nitrification inhibitors on soil nitrification, maize yield and nitrogen use efficiency in three agricultural soils with pot experiment. Compared to the other studies, this research is not novel. Other issues that need revision or clarification include:

(1) In introduction: It mentioned in line 67-68 that "there has been little research on the effect of NIs combinedwith ammonium sulfate in soils". However, in this study, they did not compare ammonium sulfate with other N fertilizers, such as urea, or NH4Cl. It's better to summarize other studies in lines 82-90.

(2)M&M section：I suggest to arrange the orders of the three soils as Cinnamon soil, Brown soil, Red soil, i.e based on soil pH, from high to low, because soil pH is an very important parameter to affect nitrification.

The available K of Brown soil in Table 1 was as low as 5.7 mg/kg. It is too low. Please check it.

(3) Result: the pixels of the Figures are too low!

(4) Discussion: The mechanism behind the results need analysis.

Line 281-282,"It is worth to mention that nitrification in red soil becomes stronger, which was according with Lu et al [31], mainly due to the tillage during the decades." It seems that the nitrification of red soil was lower than the two other soils.

6. PLOS authors have the option to publish the peer review history of their article (what does this mean?). If published, this will include your full peer review and any attached files.

Reviewer #1: No

Reviewer #2: No

Reviewer #3: **Yes: **Junhong Bai

Reviewer #4: No

---

## [Author Response · Author response to Decision Letter 0]

18 Aug 2021

Response to academic editor’s comments

We ensured that our manuscript meets PLOS ONE's style requirements, including those for file naming.

We note that one or more of the authors are employed by a commercial company: North Huajin Chemical Industries Group Corporation, Jinxi Natural Gas Chemical Co. Ltd.

We ensured that we have specifically and accurately indicated the role(s) that these authors had in our study. We confirmed that this commercial affiliation does not alter your adherence to all PLOS ONE policies on sharing data and materials by including the following statement.

3. PLOS requires an ORCID iD for the corresponding author in Editorial Manager on papers submitted after December 6th, 2016. Please ensure that you have an ORCID iD and that it is validated in Editorial Manager. 

We ensured that we have an ORCID iD and that it is validated in Editorial Manager. 

Response to reviewers’ comments

Reviewer #1: This manuscript described the effects of different nitrification inhibitors on soil nitrification and maize yield in various agricultural soils with different physicochemical properties. The subject is interesting and the paper is well organized. The introduction gave a satisfactory literature survey. The research methods used in this paper were described in detail and the data was well statistically analyzed. Appropriate figures were given to make the paper understood easily.

From my point of view, the work is well-done and provides theoretical basis to utilize fertilizer nitrogen effectively on maize in three soils with different nitrification inhibitors. The paper conclusions provided critical support for the theory that different nitrification inhibitors have different effects in different physical and chemical properties of soils. In addition, it seeks to find an economical way to improve NUE with new nitrification inhibitor N-GD due to higher price of DMPP. The results also showed that soil pH plays an important role in the effectiveness of DMPP, N-GD and their combination in influencing soil nitrification, yield and NUE of maize. Therefore, this article deserves to be published.

Some minor modifications should be made before publication:

RESPONSE: We appreciate the suggestions from the reviewer, which help us greatly improve our manuscript.

1. In this manuscript, "decrease nitrification" should be revised as “inhibit/suppress nitrification".

It was revised.

2. Page 2, line 24, please revise “The application” to “ Application”.

It was revised.

3. Page 2, line 25, please revise “the most efficiency” to “the most efficient”.

It was revised.

4. Page 2, line 38, please add “that” after “than”.

It was added.

5. Page 3, line 59, please delete “which”.

It was deleted.

6. Page 4, line 78, please add “and” in front of “field”.

It was added.

7. Page 4, line 82, please add parentheses.

It was added.

8. Page 5, line 90, please delete “are”.

It was deleted.

9. Page 7, line 139, please revise “sampling” to “samples”.

It was revised.

10. Page 11, line 216, Page 16, line 332, please revise “was” to “were”.

It was revised.

11. Page 17, line 341, please add “of maize” after “NUE”.

It was added.

12. Page 17, line 343, “improving” should be revised to “improve”.

It was revised.

Reviewer #2: To understand the efficiency of nitrification inhibitors with N fertilizer on soil nitrification, yield and nitrogen use efficiency of maize (Zea mays L.), an outdoor pot experiment with different nitrification inhibitors in three soils with different pH were conducted in the paper. The reviewer think that the problems of this paper are as follows:

RESPONSE: We appreciate the suggestions from the reviewer, which help us greatly improve our manuscript.

1. There are some sentences with irregular grammar or incomplete meaning: Such as:

(1) L39:”DMPP and N-GD were more effective in the red soil than in the cinnamon soil” What is more effective?

After careful checking, it was changed “DMPP was more effective in the red soil than in the cinnamon soil”. 

(2)L40 and 41:“… than other two soils, … than other two soils” were changed to“…than that in other two soils,… than that in other two soils.”

It was changed.

(3)L169:“NH4+-N content and NO3--N content” should be changed to“Contents of NH4+-N and NO3--N in soils”.

It was changed.

(4)L197:“Apparent nitrification rate (ANR) indicates that intensify of soil nitrification”, the sentence is not a clause, it should be changed to “Apparent nitrification rate (ANR) indicates the intensify of soil nitrification”.

It was changed.

(5)L253:“Agronomy nitrogen use efficiency and apparent nitrogen recovery” should be changed to“Utilization efficiency of agronomy ang recovery of apparent nitrogen”.

It was changed.

(6)L327:“was more stable in lower pH soil than in alkaline soil” should be changed to“was more stable in lower pH soil than that in alkaline soil”.

It was changed.

2. Improper expression or incomplete meaning of some section titles:

(1)L146:“Analytical procedures” should be changed to “Analytical methods” (Only the determination methods are given, and there are no determination procedures or operation processes)

It was changed.

(2)Note: To distinguish the relationship between the instrument and the method, the instrument only is the means of the method, but the instrument itself is not the method. Such as: “with a pH meter” in L147, “with an elemental analyzer” in L148, “determined by the flame photometer method” in L150-151, and “using a Continuous Flow Analyzer” in L151-152. Above all, these sentences should be expressed in the form of “methods” rather than “instruments”, especially, “determined by the flame photometer method” in L150-151 was a wrong expression, should be changed to “determined by the flame photometry method”.

We have rewrite these sentences to express in the form of “methods”. It was changed.

(3) The meaning of “Apparent nitrification rate” in L196 was too general.

We revised the meaning of “Apparent nitrification rate” to make it more specific.

3. Some tense usages were not correct, the past tense should be used in many sentences but used the present tense.

It was corrected.

4. In the description of result and content discussion, the first person is used too much. Such as: "we conduct an outdoor.." in L99, “We clearly observed that…..” in L170(Figure 1 should be described first), “We also found that…” in L179, “we clearly observed……” in L340(It is more appropriate to have such a statement in the conclusion). It is suggested that the result or content described in the first person should be described in the passive voice.

We fully agreed with the suggestion of the reviewer. According the suggestions, we described in the passive voice to describe the result or content described in the first person. Such as “an outdoor… was conducted” . The first person have been deleted. In the conclusion, we deleted the sentence “we clearly observed……”.

5. “In our study” is used too often, which is to discuss the results of this paper, so it is unnecessary to emphasize it many times.

We fully agree with the suggestion of the reviewer. According the suggestions, we deleted “In our study”.

6. The expression of this comparison "The results was similarly to our study,....."in L332 is improper, it should be expressed "The results of this paper was similarly to that of the literature,.....".

It was changed.

7. The part of “conclusion” is an objective statement of the law presented by the research results, The expression of “In our study, we clearly observed” should be deleted.

We deleted the expression of “In our study, we clearly observed”.

8. The logical relationship between several experimental results in the result part is not very clear, moreover, the logical relationship between the results and the discussion is also fuzzy.

We reorganized the results and the discussion to make the content clear.

Reviewer #3: Opinion on manuscript: PONE-D-21-09541 major revision

Global food security is a key challenge of the modern world. The demand for food is growing exponentially and in this regard agricultural production has to be increased to meet global food demands. One of the key agricultural practices is the higher application of nitrogenous fertilizers. However, the use efficiency of applied fertilizers is currently very low. This low efficiency is attributed to the great losses of nitrogenous fertilizers after their application into the soil due to leaching, run off, volatilization and de-nitrification. Overall, the findings of this study have essential implications for the enhancing NUE and apparent nitrogen recovery. The comments are as follows, and most of them are suggestions to the authors that should be taken on board for a revised version.

RESPONSE: We appreciate the suggestions from the reviewer, which help us greatly improve our manuscript.

Specific comments:

1. Page 2 - line 25 –“efficiency ways” should be changed into “efficient ways”

It was changed.

2. Page 2 - line 32 –“which decreased apparent nitrification rate 28%” should be changed into “which decreased apparent nitrification rate by 28%”

It was changed.

3. Page 2 - line 35 –“5.07 times” delete “times”

It was deleted.

4. Page 4 - line 67,69, 88 – “But” “And” should not be the first word of the sentence and rewrite this.

We have already rewritten this.

5. Page 4 - line 81 – “A laboratory incubation experiment…” should be marked with a reference.

We have already marked the reference.

6. Page 8 - line 147 – “pH meter” should be marked with the mode.

We have already marked the mode.

7. Page 8 – line 149 – “molybdenum blue method” reference?

The reference has been marked. 

8. Page 10 - line 197 – “Apparent nitrification rate” should be abbreviated.

We abbreviated “Apparent nitrification rate” as “ANR”.

9. Page 13 - line 255 – “Agronomy nitrogen use efficiency” should be changed into “agronomic nitrogen use efficiency”.

It was changed “Agronomic nitrogen use efficiency”. 

10. Page 14 - line 289 – “which was line in with” Grammatical errors

It was corrected.

10. Page 15 - line 295-297 the discussion is so simple, rewrite this part.

We have rewritten this part. 

10. Page 14 - line 314 – “environmental factors, soil properties and the type of fertilizer” should be more detailed. Rewrite this part according to the results of this article.

We have rewritten this part according to the results of this article.

10. Page 16- line 323 – “In our study, the efficiency of NIs differed in different agricultural soils, mainly due to the different pH”. The conclusion is far fetched and rewrite this sentence.

The effects of pH on nitrification should be more detailed.

We have rewritten this sentence. The effect of pH on nitrification has been described in detail.

11. Page 16- line 322 – Please add a “Table for correlation analysis between soil properties and NIs use” if possible.

The application of NIs have the potential to slow soil nitrification, thus affecting soil inorganic (NH4+-N and NO3--N) concentrations. Therefore, we added a “Table for correlation analysis among soil properties, NH4+-N and NO3--N”.

12. please calculate the nitrification inhibition rate of NIs

We already calculated the nitrification inhibition rate of NIs, and added a table in the supplementary material.

13. The whole manuscript some of the expressions are too colloquial and the meaning is not clearly expressed, please check the grammar in the article carefully.

We revised some of the expression of the manuscript to make the content clear. We checked the grammar in the article carefully, and corrected it. 

Reviewer #4: This manuscript had compared the effects of combined nitrification inhibitors on soil nitrification, maize yield and nitrogen use efficiency in three agricultural soils with pot experiment. Compared to the other studies, this research is not novel. Other issues that need revision or clarification include:

RESPONSE: We appreciate the suggestions from the reviewer, which help us greatly improve our manuscript. The paper aims to compare the different effects of nitrification inhibitors in different physical and chemical properties of soils, and this is the first research to study the effect of new nitrification inhibitor (nitrogen protectant (N-GD)) in different soils. 

(1) In introduction: It mentioned in line 67-68 that "there has been little research on the effect of NIs combined with ammonium sulfate in soils". However, in this study, they did not compare ammonium sulfate with other N fertilizers, such as urea, or NH4Cl. It's better to summarize other studies in lines 82-90.

We summarized other studies in the revised manuscript.

(2)M&M section：I suggest to arrange the orders of the three soils as Cinnamon soil, Brown soil, Red soil, i.e based on soil pH, from high to low, because soil pH is an very important parameter to affect nitrification.

The available K of Brown soil in Table 1 was as low as 5.7 mg/kg. It is too low. Please check it.

We fully agreed with the suggestion of the reviewer. We arranged the orders of the three soils as Cinnamon soil, Brown soil, Red soil. We checked the available K of brown soil, and corrected it.

(3) Result: the pixels of the Figures are too low!

We adjusted the pixels of the Figures.

(4) Discussion: The mechanism behind the results need analysis.

Line 281-282,"It is worth to mention that nitrification in red soil becomes stronger, which was according with Lu et al [31], mainly due to the tillage during the decades." It seems that the nitrification of red soil was lower than the two other soils.

It is well known that nitrification in natural red soil is extremely weak due to low pH and soil texture, but nitrification has been accelerated in many such soils following tillage in recent decades. However, the nitrification of red soil was lower than the two other soils. The reason of this phenomenon is its lower pH and soil organic matter.

---

## [Decision Letter · Decision Letter 1]

5 Oct 2021

PONE-D-21-09541R1Effects of combined nitrification inhibitors on soil nitrification, maize yield and nitrogen use efficiency in three agricultural soilsPLOS ONE

Dear Dr. Li,

Thank you for submitting your manuscript to PLOS ONE. After careful consideration, we feel that it has merit but does not fully meet PLOS ONE’s publication criteria as it currently stands. Therefore, we invite you to submit a revised version of the manuscript that addresses the points raised during the review process.

We look forward to receiving your revised manuscript.

Kind regards,

Manuel Joaquín Reigosa, Ph.D.

Academic Editor

PLOS ONE

Journal Requirements:

Reviewers' comments:

Reviewer's Responses to Questions

**Comments to the Author**

1. If the authors have adequately addressed your comments raised in a previous round of review and you feel that this manuscript is now acceptable for publication, you may indicate that here to bypass the “Comments to the Author” section, enter your conflict of interest statement in the “Confidential to Editor” section, and submit your "Accept" recommendation.

Reviewer #1: All comments have been addressed

Reviewer #2: All comments have been addressed

Reviewer #3: All comments have been addressed

2. Is the manuscript technically sound, and do the data support the conclusions?

Reviewer #1: Yes

Reviewer #2: Yes

Reviewer #3: Yes

3. Has the statistical analysis been performed appropriately and rigorously? 

Reviewer #1: Yes

Reviewer #2: Yes

Reviewer #3: Yes

4. Have the authors made all data underlying the findings in their manuscript fully available?

Reviewer #1: Yes

Reviewer #2: Yes

Reviewer #3: Yes

5. Is the manuscript presented in an intelligible fashion and written in standard English?

Reviewer #1: Yes

Reviewer #2: Yes

Reviewer #3: Yes

6. Review Comments to the Author

Reviewer #1: I am happy to see that the authors have answered all the queries raised by the reviewers. After revision the MS is scientifically more sound than the first submitted form. In my opinion the MS is fit for publication.

Reviewer #2: The manuscript has been carefully revised by the author, but there are still several small problems as follows：

1. L26 and L102,103: The active voice such as “we conducted an outdoor pot ……” used in the abstract L26 is inconsistent with the passive voice such as “an outdoor pot experiment with different types of NIs additions in the above three soils was conducted” in the preface L102,103. It suggested that the passive voice should be used as much as possible in the full text.

2.L65，67: There are no percentage signs for some quantities such as “58.5 in L65” and “83.8” that need to be added with a percentage sign “%”, i.e. change 58.5 to 58.5% in L65 and change 83.8 to 83.8% in L67.

3. The meaning of some percentages such as “93.5% (pH: 7.0, 93.5%), 85.1% (pH: 8.0, 85.1%) and 70.5% (pH: 4.6, 70.5%) in L83 and 84 is not clear.

4. To make the meaning of the sentence of “Samples were shaken for 1 h on a reciprocal shaker, filtered and the extract analyzed on a Continuous Flow Analyzer “ clearer (i.e. the contents of NH4+-N and NO3—N in the extract of soil were analyzed on a Continuous Flow Analyzer), the two sentences “The soil NH4+-N and NO3—N content were determined by extracting a 5-g soil subsamples with 50 ml of 2 mol L-1 potassium chloride (KCl)” in L157 and “Samples were shaken for 1 h on a reciprocal shaker, filtered and the extract analyzed on a Continuous Flow Analyzer” in L158 should be combined into one sentence, for example, change the two sentences to “The soil NH4+-N and NO3—N content were determined by extracting a 5-g soil subsamples with 50 ml of 2 mol L-1 potassium chloride (KCl) and the samples were shaken for 1 h on a reciprocal shaker, filtered and the extract analyzed on a Continuous Flow Analyzer”.

5.L161：Is the concept “Apparent nitrification rate” in line 161 mentioned for the first time? If this concept is mentioned for the first time, the significance of calculating “Apparent nitrification rate” should be introduced briefly. Otherwise, the reader does not know why “Apparent nitrification rate” should be calculated in this paper.

6. The abbreviation “ANR” of “Apparent nitrification rate” should be clearly marked when it appears for the first time in L204. Moreover, what the reviewers don't understand is that the “Apparent nitrification rate” in the first sentence in L204 is abbreviated, but why does the “Apparent nitrification rate” in the second sentence in L204 and 205 use the full name?

Reviewer #3: This manuscript describes the effects of nitrification inhibitors on nitrification and maize yield in different agricultural soils, which is a very meaningful study. The authors have adequately addressed the comments raised in a previous round of review and this manuscript is now acceptable for publication. The manuscript has described a technically sound piece of scientific research with data that supports the conclusions, and the statistical analysis has been performed appropriately and rigorously. In my opinion, this work is well written and provides a theoretical basis for utilization. Therefore, I suggest that the article could be accepted and published in this journal.

7. PLOS authors have the option to publish the peer review history of their article (what does this mean?). If published, this will include your full peer review and any attached files.

Reviewer #1: No

Reviewer #2: No

Reviewer #3: No

---

## [Author Response · Author response to Decision Letter 1]

20 Oct 2021

Response to academic editor’s comments

We have reviewed the reference list to ensure that it is complete and correct.

Response to reviewers’ comments

Reviewer #1: I am happy to see that the authors have answered all the queries raised by the reviewers. After revision the MS is scientifically more sound than the first submitted form. In my opinion the MS is fit for publication.

RESPONSE: Thank you for your approval.

Reviewer #2: The manuscript has been carefully revised by the author, but there are still several small problems as follows：

RESPONSE: We appreciate the suggestions from the reviewer, which help us greatly improve our manuscript. The specific responses to the comments are as follows.

1. L26 and L102,103: The active voice such as “we conducted an outdoor pot ……” used in the abstract L26 is inconsistent with the passive voice such as “an outdoor pot experiment with different types of NIs additions in the above three soils was conducted” in the preface L102,103. It suggested that the passive voice should be used as much as possible in the full text. 

We fully agreed with the suggestion of the reviewer. According to the suggestions, we described in the passive voice in the abstract L26.

2. L65，67: There are no percentage signs for some quantities such as “58.5 in L65” and “83.8” that need to be added with a percentage sign “%”, i.e. change 58.5 to 58.5% in L65 and change 83.8 to 83.8% in L67.

It was added.

3. The meaning of some percentages such as “93.5% (pH: 7.0, 93.5%), 85.1% (pH: 8.0, 85.1%) and 70.5% (pH: 4.6, 70.5%) in L83 and 84 is not clear. 

We have rewritten the sentence to make the content clearer. The meaning of “93.5%”, “85.1%” and “70.5%” was the nitrification inhibition rate in the neutral soil (pH: 7.0), alkaline soil (pH: 8.0) and acid soil (pH: 4.5), respectively.

4. To make the meaning of the sentence of “Samples were shaken for 1 h on a reciprocal shaker, filtered and the extract analyzed on a Continuous Flow Analyzer “ clearer (i.e. the contents of NH4+-N and NO3--N in the extract of soil were analyzed on a Continuous Flow Analyzer), the two sentences “The soil NH4+-N and NO3--N content were determined by extracting a 5-g soil subsamples with 50 ml of 2 mol L-1 potassium chloride (KCl)” in L157 and “Samples were shaken for 1 h on a reciprocal shaker, filtered and the extract analyzed on a Continuous Flow Analyzer” in L158 should be combined into one sentence, for example, change the two sentences to “The soil NH4+-N and NO3--N content were determined by extracting a 5-g soil subsamples with 50 ml of 2 mol L-1 potassium chloride (KCl) and the samples were shaken for 1 h on a reciprocal shaker, filtered and the extract analyzed on a Continuous Flow Analyzer”. 

We fully agreed with the suggestion of the reviewer. We have combined the two sentences into one sentence.

5.L161：Is the concept “Apparent nitrification rate” in line 161 mentioned for the first time? If this concept is mentioned for the first time, the significance of calculating “Apparent nitrification rate” should be introduced briefly. Otherwise, the reader does not know why “Apparent nitrification rate” should be calculated in this paper.

We have introduced the significance of calculating “Apparent nitrification rate”briefly.

6. The abbreviation “ANR” of “Apparent nitrification rate” should be clearly marked when it appears for the first time in L204. Moreover, what the reviewers don't understand is that the “Apparent nitrification rate” in the first sentence in L204 is abbreviated, but why does the “Apparent nitrification rate” in the second sentence in L204 and 205 use the full name?

To make the contents clearer, we have changed the abbreviation “ANR” to “Apparent nitrification rate” when it appears for the first time. Moreover, the abbreviation “ANR” of “Apparent nitrification rate” was used in the second sentence.

Reviewer #3: This manuscript describes the effects of nitrification inhibitors on nitrification and maize yield in different agricultural soils, which is a very meaningful study. The authors have adequately addressed the comments raised in a previous round of review and this manuscript is now acceptable for publication. The manuscript has described a technically sound piece of scientific research with data that supports the conclusions, and the statistical analysis has been performed appropriately and rigorously. In my opinion, this work is well written and provides a theoretical basis for utilization. Therefore, I suggest that the article could be accepted and published in this journal.

RESPONSE: Thank you for your approval.

---

## [Decision Letter · Decision Letter 2]

23 Feb 2022

PONE-D-21-09541R2Effects of combined nitrification inhibitors on soil nitrification, maize yield and nitrogen use efficiency in three agricultural soilsPLOS ONE

Dear Dr. Li,

Thank you for submitting your manuscript to PLOS ONE. After careful consideration, we feel that it has merit but does not fully meet PLOS ONE’s publication criteria as it currently stands. Therefore, we invite you to submit a revised version of the manuscript that addresses the points raised during the review process.

We look forward to receiving your revised manuscript.

Kind regards,

Manuel Joaquín Reigosa, Ph.D.

Academic Editor

PLOS ONE

Journal Requirements:

Reviewers' comments:

Reviewer's Responses to Questions

**Comments to the Author**

1. If the authors have adequately addressed your comments raised in a previous round of review and you feel that this manuscript is now acceptable for publication, you may indicate that here to bypass the “Comments to the Author” section, enter your conflict of interest statement in the “Confidential to Editor” section, and submit your "Accept" recommendation.

Reviewer #1: All comments have been addressed

Reviewer #2: All comments have been addressed

Reviewer #3: All comments have been addressed

2. Is the manuscript technically sound, and do the data support the conclusions?

Reviewer #1: Yes

Reviewer #2: Yes

Reviewer #3: Yes

3. Has the statistical analysis been performed appropriately and rigorously? 

Reviewer #1: Yes

Reviewer #2: Yes

Reviewer #3: Yes

4. Have the authors made all data underlying the findings in their manuscript fully available?

Reviewer #1: Yes

Reviewer #2: Yes

Reviewer #3: (No Response)

5. Is the manuscript presented in an intelligible fashion and written in standard English?

Reviewer #1: Yes

Reviewer #2: No

Reviewer #3: (No Response)

6. Review Comments to the Author

Reviewer #1: The authors have made detailed modifications according to the comments of reviewers, and I think it is now suitable for publication in this journal.

Reviewer #2: Comments on contents：

1. In this study, three soils with different pH were used for pot experiment, which showed that the biggest difference in physical and chemical properties of three different types of soils was pH. However, the conclusion showed that pH was the main factor affecting the utilization rate of three soil nitrification inhibitors without discussing the impact of other soil properties, this conclusion was preconceived and far fetched. Is the conclusion credible?

2. In Table 1, except for soil pH, other factors are the total amount and available content of nutrient elements affected by soil properties, which cannot be used as soil properties.

3. In “materials and methods”: soil sample collection methods are described in the lines of 105 and 139. What is the relationship or difference between them in L105 and 139? It is suggested to combine these two parts related to soil sample collection.

4. L345 and 346：As one of conclusions “The study shows that the effect of different NIs varied among different soils, and soil pH is the main factor.” The conclusion is too far fetched, and the statement of the conclusion is too vague and the content is not clear.

5. L80，81: “nitrification is rapid in soils of pH ≥ 6.0, but slower in soils of pH ≤ 5.0”(Is there a quantitative calculation to express speed in terms of rate? It can be expressed as easy or not)

Comments on grammatical errors or nonstandard expressions:

1. L169: It suggested that the title “NH4 + -N content and NO 3 - -N content” should be changed to “The contents of NH4 + -N and NO 3 - -N in soils”

2. It is not necessary to emphasize this research or frequently express research results or opinions in the first person during the results and discussion, otherwise the subjectivity of scientific research papers is too strong and does not accord with the objectivity of scientific papers. For example：

L27: “ In the present research”

L75: “Our study therefore addresses”

L170：“We clearly observed“

L179: “We also found that”

L294：“We also found that at the……”

L313: “which was in contrast to our results”

L320：“thus our best choice”,

L323 and 326：“In our study”

L332：“The results was similarly to our study”

L340: “In our study, we clearly observed”

L345：“The study shows that”(The conclusion is of course the conclusion of this study, and this emphasis has no significance at all)

3. The sentences in L47 and 48 should be combined into compound sentences(“asignificantly contribution in alleviating the global food shortage. Hence, tons of N fertilizers arepplied to obtain high yield of grain”)

4. Grammer irregularities or errors:

L59: “Two NIs, 3,4-dimethylpyrazol phosphate (DMPP) and dicyandiamide (DCD), which are commonly used” (There is no predicate, and the clause guided by which is redundan)

L77: “Different NIs have different effects mainly due to the properties of NIs,”(What is the impact, Unclear content description)

L310: “Ammonium ions usually adsorb by clay particles or soil organic matter, thus decreasing N loss”(It's the passive voice, not the active voice)

L312: “However, the results of Huérfano et al and Guardia et al showed that there are no any effect of“ (The present tense should be the past tense)

L320-321: “thus our best choice is the application of N-GD and DMPP+N-GD in brown soil and cinnamon soil, respectively.”

L330: “the effectiveness of NIs also influenced by SOM”

5. Nonstandard expressions：

L64，66，67: “58.5 and 35.2%” , “0.3-41.1% a” , “6.3-34.4%” should be changed into “58.5% and 35.2%” , “0.3%-41.1% a” , “6.3%-34.4%”.

6. Unclear or confused logical relationship:

L87 and 88：“but in grain yield significantly higher than those of control treatment,”“ however, there was no significant difference between NIs and without NIs [22]. But another field……”(but……, however……… but….. How to understand these three continuous turning relationships? )

L281 and 282：It is worth to mention that nitrification in red soil becomes stronger, which was according with Lu et al [31], mainly due to the tillage during the decades(How to understand the relationship between “which was according with……., mainly due to ” ).

Reviewer #3: (No Response)

7. PLOS authors have the option to publish the peer review history of their article (what does this mean?). If published, this will include your full peer review and any attached files.

Reviewer #1: No

Reviewer #2: No

Reviewer #3: No

---

## [Author Response · Author response to Decision Letter 2]

11 Mar 2022

Response to academic editor’s comments

We have reviewed the reference list to ensure that it is complete and correct.

Response to reviewers’ comments

Reviewer #1: The authors have made detailed modifications according to the comments of reviewers, and I think it is now suitable for publication in this journal.

RESPONSE: Thank you for your approval.

Reviewer #2:

1. In this study, three soils with different pH were used for pot experiment, which showed that the biggest difference in physical and chemical properties of three different types of soils was pH. However, the conclusion showed that pH was the main factor affecting the utilization rate of three soil nitrification inhibitors without discussing the impact of other soil properties, this conclusion was preconceived and far fetched. Is the conclusion credible?

Thank you for your review. We have rewritten the conclusion to make it credible. In the discussion part, except for soil pH, we also have discussed the impact of soil organic matter (SOM) on the efficiency of nitrification inhibitors. Nitrification inhibitors are compounds that delay the process of nitrification of NH4+ to NO3-. Therefore, Pearson correlation analysis between soil pH, soil organic matter (SOM), NH4+-N and NO3--N was used to further confirmed our conclusion. 

2. In Table 1, except for soil pH, other factors are the total amount and available content of nutrient elements affected by soil properties, which cannot be used as soil properties.

Thank you for your review. Generally speaking, the soil chemical properties refer to the chemical composition of soil, including the content of various elements in the soil. In addition, many literatures consider the content of nutrient elements in soil as soil properties.

3. In “materials and methods”: soil sample collection methods are described in the lines of 105 and 139. What is the relationship or difference between them in L105 and 139? It is suggested to combine these two parts related to soil sample collection. 

Thank you for your review. Line 105 describes the tested soil in this study, while L139 is the collection method of soil samples for analysis in each growth period of pot experiment. We have changed “Study site and soil sampling” in L105 to “Study site and soils” to more clearly distinguish between them. 

4. L345 and 346：As one of conclusions “The study shows that the effect of different NIs varied among different soils, and soil pH is the main factor.” The conclusion is too far fetched, and the statement of the conclusion is too vague and the content is not clear.

Thank you for your review. We have rewritten the conclusion“soil pH and SOM are the main factors affecting the efficiency of NIs”to make the content clearer. Moreover, we have discussed the impact of soil pH and soil organic matter (SOM) on the efficiency of nitrification inhibitors to make the conclusion credible.

5. L80，81: “nitrification is rapid in soils of pH ≥ 6.0, but slower in soils of pH ≤ 5.0”(Is there a quantitative calculation to express speed in terms of rate? It can be expressed as easy or not)

Thank you for your review. This is a general conclusion based on the literature cited, so there is not quantitative calculation to express speed in terms of rate. We have changed the expressed from “nitrification is rapid in soils of pH ≥ 6.0, but slower in soils of pH ≤ 5.0” to “nitrification is easy in soils of pH ≥ 6.0, but not in soils of pH ≤ 5.0”.

6. L169: It suggested that the title “NH4+-N content and NO3--N content” should be changed to “The contents of NH4+-N and NO3--N in soils”

Thank you for your review. We have changed.

7. It is not necessary to emphasize this research or frequently express research results or opinions in the first person during the results and discussion, otherwise the subjectivity of scientific research papers is too strong and does not accord with the objectivity of scientific papers. For example：

L27: “ In the present research”

L75: “Our study therefore addresses” 

L170: “We clearly observed“

L179: “We also found that”

L294：“We also found that at the……”

L313: “which was in contrast to our results”

L320：“thus our best choice”,

L323 and 326：“In our study”

L332：“The results was similarly to our study”

L340: “In our study, we clearly observed”

L345：“The study shows that”(The conclusion is of course the conclusion of this study, and this emphasis has no significance at all)

Thank you for your review. We fully agreed with the suggestion of the reviewer. We have deleted them and expressed research results or opinions in the third person during the results and discussion.

8. The sentences in L47 and 48 should be combined into compound sentences(“a significantly contribution in alleviating the global food shortage. Hence, tons of N fertilizers are applied to obtain high yield of grain”)

Thank you for your review. We have combined the sentences in L47 and L48 into compound sentences (“Hence, tons of N fertilizers are applied to obtain high yield of grain, which made a significant contribution in alleviating the global food shortage”). 

9. Grammer irregularities or errors:

L59: “Two NIs, 3,4-dimethylpyrazol phosphate (DMPP) and dicyandiamide (DCD), which are commonly used” (There is no predicate, and the clause guided by which is redundan)

L77: “Different NIs have different effects mainly due to the properties of NIs,”(What is the impact, Unclear content description) 

L310: “Ammonium ions usually adsorb by clay particles or soil organic matter, thus decreasing N loss”(It's the passive voice, not the active voice)

L312: “However, the results of Huérfano et al and Guardia et al showed that there are no any effect of“ (The present tense should be the past tense)

L320-321: “thus our best choice is the application of N-GD and DMPP+N-GD in brown soil and cinnamon soil, respectively.”

L330: “the effectiveness of NIs also influenced by SOM”

Thank you for your review. L59: It was changed to “Two commonly used NIs are 3,4-dimethylpyrazol phosphate (DMPP) and dicyandiamide (DCD)”.

L77: It was changed to “The main factors affecting the effect of NIs are the properties of NIs,” to make the content clearer.

L310: It was changed the passive voice.

L312: It was changed 

L320-321: It was changed to “thus the best choice in the brown soil and cinnamon soil is the application of N-GD and DMPP+N-GD, respectively.” to make the content clearer.

L330: It was changed to “SOM also affected the effectiveness of NIs”.

10. Nonstandard expressions：

L64，66，67: “58.5 and 35.2%” , “0.3-41.1% a” , “6.3-34.4%” should be changed into “58.5% and 35.2%” , “0.3%-41.1% a” , “6.3%-34.4%”.

Thank you for your review. It was changed in the revised manuscript.

11. Unclear or confused logical relationship:

L87 and 88：“but in grain yield significantly higher than those of control treatment,”“ however, there was no significant difference between NIs and without NIs [22]. But another field……”(but……, however……… but….. How to understand these three continuous turning relationships? )

L281 and 282：It is worth to mention that nitrification in red soil becomes stronger, which was according with Lu et al [31], mainly due to the tillage during the decades(How to understand the relationship between “which was according with……., mainly due to ” ).

Thank you for your review. L87 and L88: It was changed in the revised manuscript to make the content clearer.

L281 and 282: To make the logical relationship clearer, it was changed to “It is worth to mention that nitrification in the red soil has become stronger after tillage in recent decades, which is according with Lu et al”.

---

## [Decision Letter · Decision Letter 3]

27 Jun 2022

PONE-D-21-09541R3Effects of combined nitrification inhibitors on soil nitrification, maize yield and nitrogen use efficiency in three agricultural soilsPLOS ONE

Dear Dr. Li,

Thank you for submitting your manuscript to PLOS ONE. After careful consideration, we feel that it has merit but does not fully meet PLOS ONE’s publication criteria as it currently stands. Therefore, we invite you to submit a revised version of the manuscript that addresses the points raised during the review process.

Details were provided in the comment section. Comments were also inserted within the test of the manuscript. The reviewed manuscript is attached.

We look forward to receiving your revised manuscript.

Kind regards,

Rafiq Islam, Ph.D.

Academic Editor

PLOS ONE

Additional Editor Comments (if provided):

While there is a scientific merit of the research, there are several concerns associated with the manuscript which need to be addressed for further evaluation. Overall, the manuscript needs a thorough editing and reviewing.

The abstract of the manuscript needs more focus especially on actual results, significant digits, and experimental design. Likewise, introduction should have appropriate citations, avoid superlative words, and add a hypothesis. Soils information presented in the introduction should go under study sites in M&M section.

In M&M section, soils nomenclature should be used properly. There is a lack of information on experimental design and analytical methodologies. The statistical analysis of data is incomplete when data were measured over time. There should be an inclusion of time dependent random factor for few measurements. Some data need to be analyzed by using 2-way analysis of variance, while data measured over time need to be analyzed by using 3-way analysis of variance for treatment, soil, time, treatment x soil, treatment x time, soil x time, and treatment x soil x time effects. Needs to reanalyzed the data.

Furthermore, how was soil moisture adjusted at 60% throughout the pot? How the soil water holding capacity was measured? Needs to explain the methodology.

Results and discussion will change as per new statistical analysis. Correlation presented in Table 3 is linear; but it needs to check whether are there any non-linear responses.

Graphs ( 1 and 2) were wrong. If growth stages are used as a qualitative predictor variable, then graphs should be prepared in bars. Line graphs for continuous variables like days (number). The authors should use days after planting (DAP) as a quantitative variable to prepare line graphs.

Reviewers' comments:

Reviewer's Responses to Questions

**Comments to the Author**

1. If the authors have adequately addressed your comments raised in a previous round of review and you feel that this manuscript is now acceptable for publication, you may indicate that here to bypass the “Comments to the Author” section, enter your conflict of interest statement in the “Confidential to Editor” section, and submit your "Accept" recommendation.

Reviewer #1: All comments have been addressed

Reviewer #2: All comments have been addressed

Reviewer #3: All comments have been addressed

2. Is the manuscript technically sound, and do the data support the conclusions?

Reviewer #1: Yes

Reviewer #2: Yes

Reviewer #3: Yes

3. Has the statistical analysis been performed appropriately and rigorously? 

Reviewer #1: Yes

Reviewer #2: Yes

Reviewer #3: Yes

4. Have the authors made all data underlying the findings in their manuscript fully available?

Reviewer #1: Yes

Reviewer #2: Yes

Reviewer #3: (No Response)

5. Is the manuscript presented in an intelligible fashion and written in standard English?

Reviewer #1: Yes

Reviewer #2: Yes

Reviewer #3: (No Response)

6. Review Comments to the Author

Reviewer #1: The authors have  addressed all my comments.  The quality of the paper has been improved, thus  I have no further comments.

Reviewer #2: 1. Pay attention to the correct use of tense: the results of the cited literature and the research results of this paper should be in the past tense.

2. It is recommended to use a three line table instead.

3. The font size of abscissa, ordinate and legend of all figures are too small to identify their meaning.

4. Abbreviations should not be used in conclusions, it is suggested that the abbreviations in the conclusion should be expressed in full to make the conclusion more clear and definite.

Reviewer #3: The authors have solved the reviewers' concerns. I have no futher comments. I recommend its accpetance for publication in this journal.

7. PLOS authors have the option to publish the peer review history of their article (what does this mean?). If published, this will include your full peer review and any attached files.

Reviewer #1: No

Reviewer #2: No

Reviewer #3: No

---

## [Author Response · Author response to Decision Letter 3]

28 Jul 2022

Response to additional editor’s comments

1．The abstract of the manuscript needs more focus especially on actual results, significant digits, and experimental design. Likewise, introduction should have appropriate citations, avoid superlative words, and add a hypothesis. Soils information presented in the introduction should go under study sites in M&M section.

Thank you for your review. We have revised abstract and added a hypothesis in the introduction. Additionally, soils information presented in the introduction have been placed in M&M section.

3. In M&M section, soils nomenclature should be used properly. There is a lack of information on experimental design and analytical methodologies. The statistical analysis of data is incomplete when data were measured over time. There should be an inclusion of time dependent random factor for few measurements. Some data need to be analyzed by using 2-way analysis of variance, while data measured over time need to be analyzed by using 3-way analysis of variance for treatment, soil, time, treatment x soil, treatment x time, soil x time, and treatment x soil x time effects. Needs to reanalyzed the data.

Thank you for your review. We have added the information on experimental design and analytical methodologies and analyzed the data measured over time by using 3-way analysis of variance.

4. Furthermore, how was soil moisture adjusted at 60% throughout the pot? How the soil water holding capacity was measured? Needs to explain the methodology.

Thank you for your review. The method of measuring soil water holding capacity is as follows: Firstly, take soil with a ring knife, bring it back to the laboratory, lay a filter paper under the ring knife, fix it on the ring knife with rubber band; secondly, put the ring knife on a plate, pour water into the plate until the filter paper is soaked, the soil in the ring knife was removed and weighed in the aluminum box of known weight to obtain the soil mass W1 at the maximum water holding capacity; thirdly, the aluminum box was dried in the oven at 105℃ for more than 8 hours to constant weight, and then weighed to obtain the dry soil mass W2; finally, the maximum water holding capacity was obtained according to the formula (W1-W2) /W2. The soil water content adjusted about 60% of the maximum water holding capacity in the field, that is, the soil water content was about 20%. The specific steps are as follows: 1) The water content of the tested soil is determined as A (%); 2) When the water content is A, the weight of the tested soil required for 8kg dry soil (the amount of soil in each pot is converted to 8kg dry soil) is calculated，which is B (kg); 3) When the water content is 20%, the weight of the tested soil required for 8kg dry soil, namely C (kg); 3) B-C refers to the weight of water added to the soil, so that the water content in each pot of soil reaches about 60% of the maximum water holding capacity in the field.

5. Results and discussion will change as per new statistical analysis. Correlation presented in Table 3 is linear; but it needs to check whether are there any non-linear responses.

Thank you for your review. We have changed the results and discussion. Additionally, we have made a linear regression, the results of linear regression are shown below, and their significance are 0.000, so there are no none-linear responses.

6. Graphs ( 1 and 2) were wrong. If growthstages are used as a qualitative predictor variable, then graphs should be prepared in bars. Line graphs for continuous variables like days (number). The authors should use days after planting (DAP) as a quantitative variable to prepare line graphs.

Thank you for your review. We have used days after planting (DAP) as a quantitative variable to prepare line graphs.

Response to reviewers’ comments

Reviewer #1: The authors have addressed all my comments. The quality of the paper has been improved, thus I have no further comments.

RESPONSE: Thank you for your approval.

Reviewer #2:

1. Pay attention to the correct use of tense: the results of the cited literature and the research results of this paper should be in the past tense.

Thank you for your review. We have corrected the use of tense.

2. It is recommended to use a three line table instead.

Thank you for your review. But in order to meet the journal's Table guidelines, we used the current table.

3. The font size of abscissa, ordinate and legend of all figures are too small to identify their meaning.

Thank you for your review. We have adjusted the font size of abscissa, ordinate and legend of all figures to identify their meaning.

4. Abbreviations should not be used in conclusions, it is suggested that the abbreviations in the conclusion should be expressed in full to make the conclusion more clear and definite.

Thank you for your review. We have expressed the abbreviations in full to make the conclusion more clear and definite.

Reviewer #3: The authors have solved the reviewers' concerns. I have no futher comments. I recommend its accpetance for publication in this journal.

RESPONSE: Thank you for your approval.

---

## [Editor Report · Decision Letter 4]

1 Aug 2022

Effects of combined nitrification inhibitors on soil nitrification, maize yield and nitrogen use efficiency in three agricultural soils

PONE-D-21-09541R4

Dear Dr. Li,

We’re pleased to inform you that your manuscript has been judged scientifically suitable for publication and will be formally accepted for publication once it meets all outstanding technical requirements.

Kind regards,

Rafiq Islam, Ph.D.

Academic Editor

PLOS ONE

Additional Editor Comments (optional):

Nomenclature of the "red soil" is missing, which needs to be added to accept the paper for publication. Moreover, the line graphs quality should be improved with high-resolution.
---

## [Editor Report · Acceptance letter]

12 Aug 2022

PONE-D-21-09541R4 

Effects of combined nitrification inhibitors on soil nitrification, maize yield and nitrogen use efficiency in three agricultural soils 

Dear Dr. Li:

I'm pleased to inform you that your manuscript has been deemed suitable for publication in PLOS ONE. Congratulations! Your manuscript is now with our production department. 

Kind regards, 

on behalf of

Dr. Rafiq Islam 

Academic Editor

PLOS ONE